# Characterization of the 1966 Camp Century Subglacial Core: A Multiscale Analysis

Catherine M. Collins[1], Nicolas Perdrial[1, 2], Pierre-Henri. Blard[3,4], Nynke Keulen[5], William C. Mahaney[6], Halley Mastro[1], Juliana Souza[1], Donna M. Rizzo[7], Yves Marrocchi[3], Paul C. Knutz[5], Paul R. Bierman[1]

[1]Rubenstein School of Environment and Natural Resources, University of Vermont, Burlington, VT, USA
[2]Department of Geography & Geosciences, University of Vermont, Burlington, VT, USA
[3]Centre de Recherches Pétrographiques et Géochimiques, Centre national de la recherche scientifique, UMR 7358, Vandœuvre-lès-Nancy 54501, France
[4]Université Libre de Bruxelles, Laboratoire de Glaciologie, ULB, Brussels, 1050, Belgium
[5]Geological Survey of Denmark and Greenland, 1350 Copenhagen, Denmark
[6]Quaternary Surveys, 26 Thornhill Ave., Thornhill, ON L4J1J4, Canada
[7]Department of Civil and Environmental Engineering, University of Vermont, Burlington, VT, USA

*Correspondence to*: Catherine Collins (catmcollins17@gmail.com)

**Abstract.** In 1966, drilling at Camp Century, Greenland, recovered 3.44 meters of subglacial material from beneath 1350
meters of ice. Although prior analysis of this material showed that the core includes glacial sediment, ice, and sediment deposited during an interglacial period, the subglacial material had never been thoroughly studied. To better characterize this material, we analyzed 26 of the 30 core samples remaining in the archive. We performed a multi-scale analysis including X-ray diffraction, micro-computed tomography (μCT), and scanning electron microscopy to delineate stratigraphic units and assign facies based on inferred depositional processes.

At the macro-scale, quantitative X-ray diffraction revealed that quartz and feldspar dominated the sediment and that there was minimal variation in relative mineral abundance between samples. Meso-scale evaluation of the frozen material, using μCT scans, showed clear variations in the stratigraphy of the core characterized by the presence of bedding, grading, and sorting. Micro-scale grain size and shape analysis, conducted using scanning electron microscopy, showed an abundance of fine-grained materials in the lower part of the core and no correspondence between grain shape parameters and sedimentary
structures. These multiscale data define 5 distinct stratigraphic units within the core based on sedimentary process; K-means clustering analysis supports this unit delineation. Our observations suggest that ice retreat uncovered the Camp Century region exposing weathered basal till (Unit 1), now covered by a remnant of basal ice or firn (Unit 2). Continued ice-free conditions led to till disruption by liquid water causing a mass movement (Unit 3) and deposition of water-worked sediment (Units 4-5). Analysis of the Camp Century subglacial material reveals a diverse stratigraphy preserved below the ice that recorded episodes
of glaciated and deglaciated conditions in northwestern Greenland. Our physical, geochemical, and mineralogic analyses illuminate the history of deposition, weathering, and sediment transport preserved under the ice and show the promise of subglacial materials to increase our knowledge of past ice sheet behavior over time.

## 1 Introduction

Understanding past ice-free times in Greenland provides additional understanding of the Greenland Ice Sheet's (GrIS) response
to warming (Gemery & López-Quirós, 2024). This goal, deciphering Greenland's paleoclimate and past ice sheet stability and
instability, has driven the ice core collection efforts since the 1950s (Bader, 1962). Deep ice coring in Greenland began in
1960 at Camp Century, a military camp in northwestern Greenland, ~200 km inland from the ice margin (Langway, 2008)
(Figure 1).

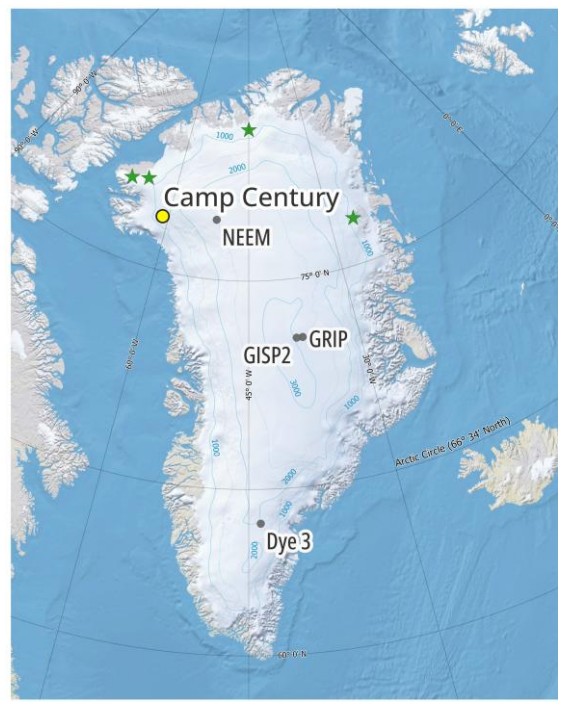

40

**Figure 1. Map of Greenland (Harrison et al., 2011; Moon et al., 2023) showing drilling locations where subglacial or basal material has been retrieved. Drilling locations for ongoing GreenDrill project (Prudhoe Land, Hiawatha Margin, Victoria Fjord, and Dronning Louise Land from West to East) are marked by a green star (Briner et al., 2022).**

At Camp Century (77.2°N, 61.1°W), the U.S. Army drilled the first deep ice core that entirely penetrated the ice sheet. Below
the ice, they collected 3.44 m of subglacial material, which would remain the longest subglacial archive for the next six decades
(Balter-Kennedy et al., 2023; Langway, 2008). Ice from this core was studied extensively (e.g., Hansen & Langway, 1966a;
Johnsen et al., 1972; Langway & Hansen, 1970; W. Dansgaard et al., 1969), but the subglacial material was left relatively
unstudied (Fountain et al., 1981; Harwood, 1986; Whalley & Langway, 1980). The subglacial material did not move to the US
Ice Core Repository with the rest of Camp Century core in the mid-1990s and was thought, by US researchers, to have been
lost. It's rediscovery in freezers of the Niels Bohr Institute at the University of Copenhagen sparked new interest in analyzing
this unique archive (Bierman et al., 2024; Voosen, 2019).

The Camp Century subglacial core is made up of several units, the deepest containing sediment last exposed to sunlight before 1.5 and buried no longer than 3.2 Myr (Bierman et al., 2024; Christ et al., 2021). Data show that the top of the subglacial core was exposed at the surface 416,000±38,000 years ago, which places its last exposure during the Marine Isotope Stage (MIS) 11 deglaciation (Christ et al., 2023). MIS 11 (374 to 424 ka) was both a long and warm interglacial with a peak atmospheric $CO_2$ concentration similar to pre-industrial levels (Dutton et al., 2015). This makes MIS 11 an important but imperfect analogue by which to study the effects of future warming and GrIS stability (Dutton et al., 2015; Jouzel et al., 2007; Lisiecki & Raymo, 2005; Lüthi et al., 2008; Robinson et al., 2017). The presence of plant and invertebrate fossils in the Camp Century subglacial core mandates the site was not covered by ice for some part of MIS 11 and indicates the maximum limit for ice extent at that time (Christ et al., 2021, 2023).

In this study, we used multiple physical, geochemical, and mineralogic techniques to characterize 26 of the 30 extant samples cut from the 3.44-meter Camp Century subglacial core (Bierman et al., 2024). We employed a multi-scale technique to define the lithological succession and analyze sedimentary facies to infer what environments were present when this sediment and ice were deposited. At the macro-scale, we conducted a mineralogical survey using X-ray diffraction (XRD) to quantify relative mineral assemblages. At the meso-scale, we used non-destructive micro-computed tomography (µCT) of the frozen sediment to make detailed stratigraphic observations at the tens of microns scale. We also used scanning electron microscopy (SEM) and associated geochemical mapping to analyze grain coatings, textures, and shapes at the micron scale. Using non-destructive µCT in tandem with other methods (XRD, SEM) allowed us to maximize data output when working with a volumetrically limited archive. Characterizing this unique subglacial core offers the opportunity to expand our knowledge of Greenland's climatic history, interglacial surface processes, and subglacial processes in addition to demonstrating the potential for subglacial materials to inform our understanding of ice sheet behavior over time.

With the fundamental core stratigraphy now defined (Bierman et al., 2024) and prior work suggesting the composition of rocks in the core (Fountain et al., 1981) as well as the presence of fossil material and the timing of interglacial exposure (Christ et al., 2021, 2023; Harwood, 1986; Whalley & Langway, 1980), the goal of this paper is to understand and document better the mineralogy of the sediment, the sedimentary sequence, the cryostructures, and the history and processes reflected by characteristics of the subglacial material.

## 2 Background

Subglacial material is a valuable source of both paleoclimate and glacial process information (e.g., Bierman et al., 2014; Schaefer et al., 2016). Sediment preserved beneath the ice and in basal ice can reveal crucial information about ice-free events, their duration, and surface processes (Bierman et al., 2024; Christ et al., 2021, 2023). Basal materials are particularly important because they contain in-situ physical evidence for past events that are not recorded in glacier ice (Bender et al., 2010; Bierman et al., 2014; Blard et al., 2023; Christ et al., 2021, 2023; Gow & Meese, 1996; Marschalek et al., 2024; Schaefer et al., 2016; Suwa et al., 2006; Yau et al., 2016).

## 2.1 Basic Ice

Nearly every deep ice core from the GrIS (Camp Century, Dye 3, Greenland Ice Sheet Project - GISP2, Greenland Ice Core Project - GRIP, North Eemian Project – NEEM, East Greenland Ice Core Project – East GRIP) has retrieved meters of deformed ice containing bed material, referred to as silty ice or basal ice (Bender et al., 2010; Blard et al., 2023; Christ et al., 2021; Gow & Meese, 1996; Hansen & Langway, 1966; Herron et al., 1979; Souchez et al., 1998; Suwa et al., 2006; Voosen, 2025). The basal ice layers (BIL), found above the ice-bed interface, are influenced by processes operating at and near the bed including deformation, reglaciation, and in some cases melting. This limits the interpretability of the climate record in the deepest ice, but the bed sediments entrained in the basal ice can reflect aspects of glacial and subglacial processes as well as paleoenvironmental conditions during ice-free episodes (Knight, 1997).

Initial analyses of debris in the Camp Century BIL confirmed, using SEM and XRD analysis, that the sediment originated from the frozen material below (Herron et al., 1979). In other locations, investigation of ancient biomolecules in Dye 3 and GRIP silty ice found evidence that a boreal forest once existed at Dye 3 which necessitates warming and extensive ice sheet retreat from southern Greenland (65.2°N), but such retreat did not appear to reach as far north as GRIP (72.5°N) (Willerslev et al., 2007). Stable water isotope composition of the BIL in the GRIP core, corroborated by a comparative study between the debris found there and at GISP2 (72.5°N), indicates that the GRIP BIL originated as ground ice before incorporation into the larger growing ice sheet and has since deformed (Gow et al., 1997; Gow & Meese, 1996; Souchez et al., 1994, 1995; Tison et al., 1994). A multiproxy analytical approach applied to sediment in the BIL from the NEEM core (77.5°N) (Blard et al., 2023) found evidence of a sequence of glacial retreat and advance possibly like that which has been identified in the Camp Century subglacial material (Christ et al., 2021, 2023).

## 2.2 Subglacial Material

In Greenland, coring activities at Camp Century, GISP2, and GreenDrill retrieved subglacial material from beneath the ice-bed interface (Balter-Kennedy et al., 2023; Christ et al., 2021; Gow & Meese, 1996; Souchez et al., 1994, 1998). At GISP2, drillers extracted 48 cm of till (mostly boulders, only 8-10 cm of fine grain material) and 1.07 m of underlying rock (Gow et al., 1997). Cosmogenic isotopic analysis of the underlying rock, till, and an overlying boulder shows that the land surface at GISP2 was deglaciated for extended periods of time during at least one and possibly many Pleistocene interglacials after the Mid-Pleistocene Transition, ~1.1 Ma (Bierman et al., 2023; Schaefer et al., 2016). Grain texture analysis of Camp Century sediments (Whalley & Langway, 1980) revealed two populations of grains mixed subglacially: angular grains, attributed to glacial crushing, and rounded grains, attributed to aeolian transport. Whalley & Langway (1980) infer that the aeolian fraction must have been produced before the ice advanced, mandating prior ice-free conditions. A petrographic investigation of 17 clasts in the Camp Century subglacial core revealed that they were compositionally like those cropping out at the ice sheet margin (Fountain et al., 1981).

Extraction of fresh-water diatoms from the bottom of the Camp Century BIL and the upper part of the sub-ice core indicates that the ice retreated far enough to expose the Camp Century drill site at least once during the Pleistocene (Harwood, 1986). Cosmogenic and luminescence dating of the upper and lowermost subglacial samples show that the 3.44 m of material records at least two glaciations separated by ice-free conditions during MIS 11 (Christ et al., 2021, 2023; Woznick, 2024). The currently active northern Greenland drilling project, GreenDrill, has successfully retrieved 3 m of silt to cobble-sized sediment in basal ice atop 4.5 m of gneissic bedrock (Balter-Kennedy et al., 2023; Briner et al., 2022).

## 2.3 Computed Tomography Stratigraphy

Computed tomography (CT), a technology widely used in the medical field, is also a powerful tool for examining sediment cores non-destructively (Renter, 1989). CT scans produce greyscale images based on the intensity of the attenuated X-ray beam and are reconstructed to produce 3-dimensional (3D) models of the scanned object (Razi et al., 2014). CT scans have been used to do quantitative facies characterization by comparing density plots that suggest environmental changes (Emmanouilidis et al., 2020; Lee et al., 2021). Used in studying marine sediment cores, CT scans aid in calculating bulk density and its spatial variation vertically and laterally (Orsi et al., 1994). CT scans also provide detailed visualizations of sediment structures, grain distribution, and material densities (Mena et al., 2015).

CT scanning technologies are effective in studying frozen permafrost cores. Some studies have characterized the sediment properties, cryostructures, and ground ice content of permafrost but did not find CT successful as a proxy for identifying ground ice origin (Calmels & Allard, 2008; Lapalme et al., 2017). µCT, a refinement of this technology, provides micron-scale resolution (typically between 1-150 µm voxels, which is the 3D equivalent to a pixel) for more detailed studies and has been used to make pore-scale observations (Lei et al., 2018, 2022). Three-dimensional microstructure analysis in permafrost allows for quantification of physical properties, including the spatial density of ice inclusions, which can be useful in estimates of thermal conductivity (Nitzbon et al., 2022). The same authors suggest that microstructure analysis from µCT scans, in combination with other measurements, allows for reliable inference of depositional processes (Nitzbon et al., 2022).

## 2.4 Microscale Grain Characteristics

Grain characteristics have been used routinely to inform sedimentological studies (Naqshband & McElroy, 2016). Grain size distribution can be an indicator of transport energy, as higher energy fluids can transport larger/heavier grains or of source material characteristics, or both (Gresina et al., 2023; Hjulström, 1935; Malusà et al., 2016). Specific grain morphology, including roundness and circularity, are related to transport energy, transport distance, mode of transport, source material, and conditions at the site of deposition (Krumbein, 1941).

Recently, studies of grain morphology have used automated image analysis (Gresina et al., 2023; Szmańda & Witkowski, 2021; Tafesse et al., 2013) to infer transport mechanisms. For example, properties of particle shape, investigated using this method, correlate circularity and roundness with transport distance and energy, respectively (Gresina et al., 2023). Image analysis of sediment grains from coastal dunes shows that roundness and patterns in size and shape indicate various and specific

modes of transport in aeolian sediments (van Hateren et al., 2020). Quartz morphology, including roundness assessments and microstructures, has revealed evidence of past storm frequency, and has been used to reconstruct sediment provenance and transport in during the Last Glacial Maximum (Kalińska-Nartiša et al., 2018; Woronko et al., 2015). Grain size analyses of loess deposits identify episodes of rapid climate change, which mimic rapid climate fluctuations during the last glacial period (Dansgaard-Oeschger events), have been found in ice cores (Vandenberghe & Nugteren, 2001). Applied to glacially derived sediments, Lepp et al. (2024) recently showed that glacial and fluvial surface textures are retained on silt-sized quartz grains and can be used to evaluate sediment transport processes.

Grain coating abundance and distribution can also provide information of source conditions, erosional processes, original transport mechanism, and post-depositional pedogenic processes. Coatings on grain surfaces indicate post-depositional weathering and pore fluid transport of solids and solutes in both temperate and Arctic regions (Dixon et al., 2002). In contrast, the absence of grain coatings may signal an active transport environment (Musselman & Tarbox, 2013). Glacial flour is common in till and glacial lacustrine deposits and is characterized by agglomerates of clay to silt-sized angular particles (Pesch et al., 2022) and small particles adhered to grain surfaces that have features consistent with glacial grinding (Mahaney, 2002; Whalley & Langway, 1980).

## 2.5 Cryostratigraphy

Cryostratigraphy describes the shape, amount, and distribution of ice and sediment in frozen ground (Gilbert et al., 2016; Murton & French, 1994). Cryofacies are defined by distinct patterns of ice lenses, volumetric ice content, and layering of ice and sediment. Cryostratigraphy is used to infer permafrost formation mechanisms as either *epigenetic*, permafrost that forms after sediment deposition, or *syngenetic*, permafrost that forms as material is deposited (French & Shur, 2010). Ground ice includes pore ice and segregated ice but typically excludes buried ice (French & Shur, 2010; Murton & French, 1994). Pore ice works as cement, holding the sediment together. Segregated ice forms as ice accumulates along the freezing plane, the boundary between conditions supporting liquid water and ice, and can be millimeters to tens of meters thick (French & Shur, 2010). Organized lenticular and layered cryostructures are common in syngenetic permafrost and tend to be short, thin, and highly abundant (French & Shur, 2010; Murton & French, 1994). Epigenetic permafrost is typically ice-poor, contains pore-filled cryostructures (Gilbert et al., 2016; Stephani et al., 2014) and is characterized by reticulate cryostructures that reflect shrinking as sediment freezes and moisture migrates toward the freezing front, a phenomenon most common in fine-grained materials (French & Shur, 2010). Thaw unconformities occur and are shown by the presence of epigenetic structures bordering diagnostic syngenetic features (French & Shur, 2010). CT scan images have been used successfully to identify cryostructures in permafrost from the McMurdo Dry Valleys of Antarctica (Lapalme et al., 2017). However, the small diameter of cores makes the identification of cryostructures uncertain due to their scale-dependence (Gilbert et al., 2016).

## 3 Methods

We employed standard geologic practices to study the stratigraphy and mineralogy of the subglacial materials. We used frozen samples for µCT, and after thawing at 4°C, we used bulk sediment for XRD and SEM analysis on a total of 26 samples from the archive. There is scant information about the storage and transportation of the samples between their collection in 1966 and resampling in 2019 (Bierman, 2024; Bierman et al., 2024; Christ et al., 2021; Voosen, 2019). The initial orientation of the core has not been preserved.

### 3.1 Micro-Computed Tomography (µCT)

We created a digital archive of the (a) and (b) sub-samples (sub-sampling described in Bierman et al. (2024), 49 in total, by collecting a series of µCT scans. We used a source voltage of 100kV, a source current of 62uA, and an exposure time of 550 ms. Correction for beam hardening was performed on a scan-by-scan basis. We scanned both sub-samples from each core section using a Bruker SkyScan1173 µCT scanner fit for use in a cold room at -10°C at the Cold Regions Research and Engineering Laboratory (CRREL) in Hanover, NH. Each of the ~10 cm-tall samples were scanned in two overlapping 7.9 cm-tall sections at a resolution of 71 µm/voxel to capture the entire length of the sample.

We completed reconstructions for each scan using the Bruker NRecon software. This resulted in 84 partial sample scans, 7 full sample scans, and 8 zoomed scans. Two scans resulted in failed reconstructions due to difficulty with the scanning procedure (sample 1063-4). We performed the same post-reconstruction manipulation for all samples in ImageJ (FIJI) (Schindelin et al., 2012): (a) we stitched the bottom and top portions of the scans (except the non-overlapping sample 1062-4) using manual calibration, b) we resliced the scans across the-z axis for visualization and c) we generated a 3D rendering of the stitched scans using a "brightest point" projection, 0% opacity, surface depth cues at 100% and interior depth cues at 50%. The raw data are archived with the Arctic Data Center (Perdrial et al., 2024). 3D visualization of all scanned samples are available for viewing and download at this online, public repository: https://www.morphosource.org/concern/cultural_heritage_objects/000583438. Qualitative assessment and 3D visualization of the partial scans, using Bruker's CTVox software and ImageJ, allowed us to investigate contacts, layering, sorting, lineations, and ice layers in each sample. The 3D nature of the CT scans allows us to examine internal structures by slicing into the models laterally. While no cross-sample calibration was performed, we were able to filter the intensity of the beam response to only display the denser suspended particles to look for other structures such as layering and particle alignment. The angularity of larger grains was also assessed visually with a classification scheme that defines grains having sharp edges as angular and grains with smooth, curved sides as rounded, and a range of intermediate shapes (Janoo, 1998).

### 3.2 X-ray Diffraction

We analyzed the crystalline composition of a representative group of 15 samples from various sections of the subglacial core (Table 1) by XRD using a Rigaku MiniFlex II, equipped with a Cu X-ray tube. Following qualitative diffractogram analysis,

we quantified mineral percentages using the Rietveld Full Pattern Profile Fit algorithm included in the PDXL-2 software (PDXL, Rigaku Corp). We used approximately 0.2 g of bulk sediment from each sample and ground it manually with a mortar and pestle. The ground sediment was mounted on a zero-background plate in random orientation mounts and analyzed in 2theta-theta geometry between 3 and 70 °2θ with a dwell time of 1 degree/min and 0.02 °2θ resolution for a total run time of 67 minutes. Although limited, the background plate contributed some background signal at low angles in the diffractogram. We characterized mineralogy using databases from the International Center for Diffraction Data 2.0 and Crystallography Open Database for peak matching. Subsequently, we performed quantitative analysis of the X-ray diffractograms using a semi-automatic Rietveld approach (Rietveld, 1969). To refine our results, we varied the values for scale factor, cell parameters (within 0.2Å), shape parameters, and for selected minerals (clays and amphiboles), the March–Dollase preferred orientation parameter, similarly to the method described in Mackowiak and Perdrial (2023). Untreated diffractograms are archived with the Arctic Data Center (Perdrial et al., 2024).

### 3.3 Scanning Electron Microscopy Imaging and Energy Dispersive X-ray Micro-mapping

We performed two different types of SEM analyses: SEM imaging and elemental mapping of polished mineral surfaces and SEM high resolution microphotography of grain textures and shapes (Table 1). We embedded thawed bulk sediments in epoxy resin (EPO-TEK 301). After curing for at least 24 hours, we micro-polished the resulting epoxy puck using a decreasing grit size to 0.05 μm. The mounts were carbon sputter-coated prior to analysis in backscattered electron (BSE) mode using a TESCAN VEGA3 scanning electron microscope coupled with an Oxford Instruments Aztec Elemental Mapping Energy Dispersive X-ray Spectrometer (EDS) in the Geology Department at Middlebury College (Vermont, USA). We acquired BSE images and EDS maps at 20 keV for a minimum of 20 frames totaling 10-minute elapsed time for each multi-elemental map. Then, we generated multi and tri-color maps using the Gatan digital micrograph 3.1 software. We imaged 15 samples with the SEM and analyzed 2 sites on each mount (except sample 1059-6 which was imaged only at one site) resulting in 29 individual images archived with the Arctic Data Center (Perdrial et al., 2024).

We conducted grain coating evaluation of all 29 images. To do so, we used the EDS maps to evaluate each grain and placed it, based on our visual assessment of grain coating abundance, into the following five unique categories: grains dominated by coating (>50% coverage), grains with moderate coating (<50% and >25% coverage), grains with little to no coating (<25% coating), grains with coating only in cracks, and fine particle aggregates. Observations from two sites of the same samples were merged so that there is one summed evaluation per sample and 5 samples were counted by two different observers to ensure consistency. Counts were converted to percentages to normalize data between samples. We created the category "High Abundance" which includes the dominated and moderately covered by coating categories to communicate abundance vs. depth more succinctly.

At the Geological Survey of Denmark and Greenland, we acquired another set of SEM images in BSE at a lower resolution for grain size and shape analysis of all mineral compositions. Images at the lower magnification were collected for 9 samples throughout the core. The 9 samples were mounted in epoxy and coated with 10 nm carbon to make them conductive.

Backscattered electron contrast (BSE) images were generated at the ZEISS Sigma 300 VP equipped with a field emission gun, using the ZEISS Mineralogic™ software platform. A mosaic of BSE frames of a representative part of the sample was taken. Further details on the software and applied method can be found in Keulen et al. (2020). Analyses were performed with acceleration voltages of 15keV using a 120 µm² aperture.

We analyzed unpolished grains from the 125 to 800 µm fractions in secondary electron mode at high resolution with SEM to observe grain microtextures. This was done at the Service Commun de Microscopie Électronique et de Microanalyse X (GeoResources, Nancy, France) using a JEOL 7600F with a 1 nA primary beam operating at 15 keV. Before collecting SEM images, grains were rinsed in milliQ water. The grains were dried and attached to adhesive paper before carbon coating. We used these high-resolution images to examine textural features at the grain surface (e.g., Cailleux & Tricart, 1959; Mahaney, 2002). We analyzed 9 samples (Table 1) and report here the key observations.

**Table 1. Summary of samples and physical, geochemical, and mineralogic analyses performed. Asterisks denote samples analyzed in work performed and reported by Christ et al. (2021).**

| Sample[a] | Depth (cm) | XRD | SEM imaging | | | µCT |
| | | | BSE, low mag[b] | BSE, high mag[c] | SE, high mag[d] | |
|---|---|---|---|---|---|---|
| 1059-4 | 0-10 | * | * | * | Analyzed | Not scanned |
| 1059-5 | 10-20 | Analyzed | Not run | Analyzed | Not run | Scanned |
| 1059-6 | 20-29.5 | Analyzed | Analyzed | Analyzed | Analyzed | Scanned |
| 1059-7 | 29.5-34 | Analyzed | Not run | Analyzed | Not run | Scanned |
| 1060-A1 | 34-44.5 | Not run | Analyzed | Not run | Analyzed | Scanned |
| 1060-A2 | 44.5-55.5 | Analyzed | Not run | Analyzed | Not run | Scanned |
| 1060-B | 55.5-78.5 | Analyzed | Analyzed | Analyzed | Not run | Not scanned |
| 1060-C1 | 78.5-88.5 | Analyzed | Analyzed | Analyzed | Analyzed | Scanned |
| 1060-C2 | 88.5-98.5 | Analyzed | Not run | Analyzed | Not run | Scanned |
| 1060-C3 | 98.5-108.5 | Not run | Analyzed | Not run | Analyzed | Scanned |
| 1060-C4 | 108.5-118 | Not run | Not run | Analyzed | Not run | Scanned |
| 1060-C5 | 118-129 | Not run | Not run | Not run | Not run | Scanned |
| 1061-A | 129-137 | Analyzed | Not run | Not run | Not run | Scanned |
| 1061-B | 137-159 | Analyzed | Not run | Analyzed | Not run | Not scanned |
| 1061-C | 159-171 | Not run | Analyzed | Not run | Not run | Scanned |
| 1061-D1 | 171-181 | Not run | Not run | Not run | Not run | Scanned |
| 1061-D2 | 181-191 | Analyzed | Not run | Not run | Not run | Scanned |
| 1061-D3 | 191-201 | Not run | Analyzed | Analyzed | Analyzed | Scanned |
| 1061-D4 | 201-215 | Missing | Missing | Missing | Missing | Missing |
| 1061-D5 | 215-223 | Analyzed | Not run | Analyzed | Analyzed | Scanned |
| 1062-1 | 223-231 | Analyzed | Not run | Analyzed | Not run | Scanned |
| 1062-2 | 231-238 | Not run | Not run | Not run | Not run | Scanned |
| 1062-3 | 238-250 | Analyzed | Not run | Analyzed | Not run | Scanned |
| 1062-4 | 250-263 | Not run | Analyzed | Not run | Not run | Scanned |
| 1063-1 | 263-273 | Not run | Not run | Not run | Not run | Not scanned |
| 1063-2 | 273-283 | Analyzed | Not run | Analyzed | Analyzed | Scanned |
| 1063-3 | 283-294.5 | Missing | Missing | Missing | Missing | Missing |

| 1063-4 | 294.5-305.5 | Not run | Not run | Not run | Not run | Not shown[e] |
| 1063-5 | 305.5-317 | Not run | Not run | Not run | Not run | Scanned |
| 1063-6 | 317-327 | Analyzed | Analyzed | Analyzed | Not run | Scanned |
| 1063-7 | 327-337 | * | * | * | Analyzed | Not scanned |

[a] Sample names notation described in (Bierman et al., 2024)

[b] BSE, low mag denotes samples analyzed in backscattered mode on polished mount, used for the morphology analysis on all particles with lower magnification.

[c] BSE, high mag denotes samples analyzed in backscattered mode on polished mount used in coating estimates (with a higher magnification), PCA and K-mean clustering.

[d] SE, high mag denotes samples analyzed in secondary electron mode used for grain texture analysis.

[e] Not shown: poor quality CT scans.

## 3.4 Image Analysis

We postprocessed the EDS images of high magnification polished grain data to determine grain size and grain shape parameters of quartz grains only. To isolate quartz grains, we used multielement color maps created with Gatan's Digital Micrograph and we performed a color threshold to select the Si-O only phases (quartz). We then performed particle analysis on the Fiji platform (Schindelin et al., 2012; Vandel et al., 2020). For this set of images, the lower limit of size for a particle to be measured was set at 150 pixels corresponding to 50 $um^2$ at the 3 pixels/um resolution we used. This minimum size is adequate for surface area and shape estimates (Francus & Pirard, 2005). We measured three grain size/shape parameters (area, roundness, and circularity) and stored the average and standard deviation for each parameter. Fiji calculates circularity as $4\pi(Area/Perimeter^2)$, which returns values between 0-1, where a value of 1.0 indicates a perfect circle. Roundness is calculated as $4*Area/(\pi*major\ axis^2)$, which returns values between 0-1, where larger values indicate increasing roundness. Areas measured by this method are lower than particle sizes obtained by physical methods (like sieving) because the arbitrary cut surface angle and depth are unlikely to correspond to a 2D projection of the particles as described in Sahagian and Proussevitch (1998).

We performed another image analysis on the set of SEM images acquired at a low magnification, allowing for a larger dataset. For grain size and shape analysis, we included all grain compositions and set the lower limit of size for a particle to be measured at 1000 $\mu m^2$. This resulted in a minimum size of 625 pixels for most images except for 2 lower magnification images (1060-C1 and 1062-4) where the smallest size particle has an area of 2440 pixels. This minimum size ensures a robust measurement of all parameters (Francus & Pirard, 2005). Grain size, circularity and roundness were measured using particle analysis in ImageJ, and a Tukey-Kramer honestly significant difference (HSD) pair-wise comparison was performed on the means for each sample in JMP Pro 15.0.0 (*JMP*, 2024). In all cases the significance level ($\alpha$) was set at 0.05. To represent significance between classes, we report differences using connecting letters where levels not connected by the same letter are significantly different. Despite the limitations of estimating the particle characteristics on polished thin sections, the use of a consistent minimum size threshold allows us to compare the parameters between samples using this method.

### 3.5 PCA Analysis and K-means Clustering

Methods for reducing the dimensionality of a dataset, including principal component analysis (PCA) and K-means clustering, can be useful when interpreting physical grain characteristics and geochemical aspects (Jansson et al., 2022). To assess our unit assignments, we performed PCA and K-means clustering on data. The size and shape parameters (including standard deviations) from the low magnification image analysis, combined with percent ice composition, percent fine-grained fraction composition, percent quartz composition, depth, and an ordinal evaluation of grain coatings varying from 0-2 (no coating, minimal coatings, and extensive coatings; determined visually) were used as input variables in a K-means clustering method on PCA variables. By integrating depth as a factor in the clustering, we follow traditional clustering stratigraphic models acknowledging depth-dependency as a deterministic factor (Gill et al., 1993; Yabe et al., 2022 ). The data set includes the 11 variables mentioned above, measured over a total of 29 observations from the 15 samples. The cluster centers were calculated using a "nearest centroid sorting" approach (Anderberg, 1973; *JMP*, 2024). We performed a PCA for dimension reduction and then performed a K-means clustering in JMP Pro 15 to assess the optimal number of clusters according to the cubic clustering criterion. *Z* scores, which normalize our input variables based on the feature mean and standard deviation, were calculated in R (R Core Team, 2022).

### 4 Results

Our multiscale analysis supports, refines, updates, and provides more justification and detail regarding unit delineations proposed earlier (Bierman et al., 2024; Christ et al., 2021, 2023; Fountain et al., 1981). PCA and subsequent K-means analysis quantitatively corroborate our understanding of these systems. Analysis of grain microtextures shows that grains throughout the subglacial core material record different histories. We find at least 16% very fine sand and silt-sized grains in all units (in some units the abundance of fines is over 90%) and more variability in grain size in the lower part of the core. The abundance of grain coatings increase with depth and coatings are largely absent in the upper two units (4 and 5). Grain shape analysis (roundness and circularity) does not vary systematically within or between units.

## 4.1 Micro-Computed Tomography

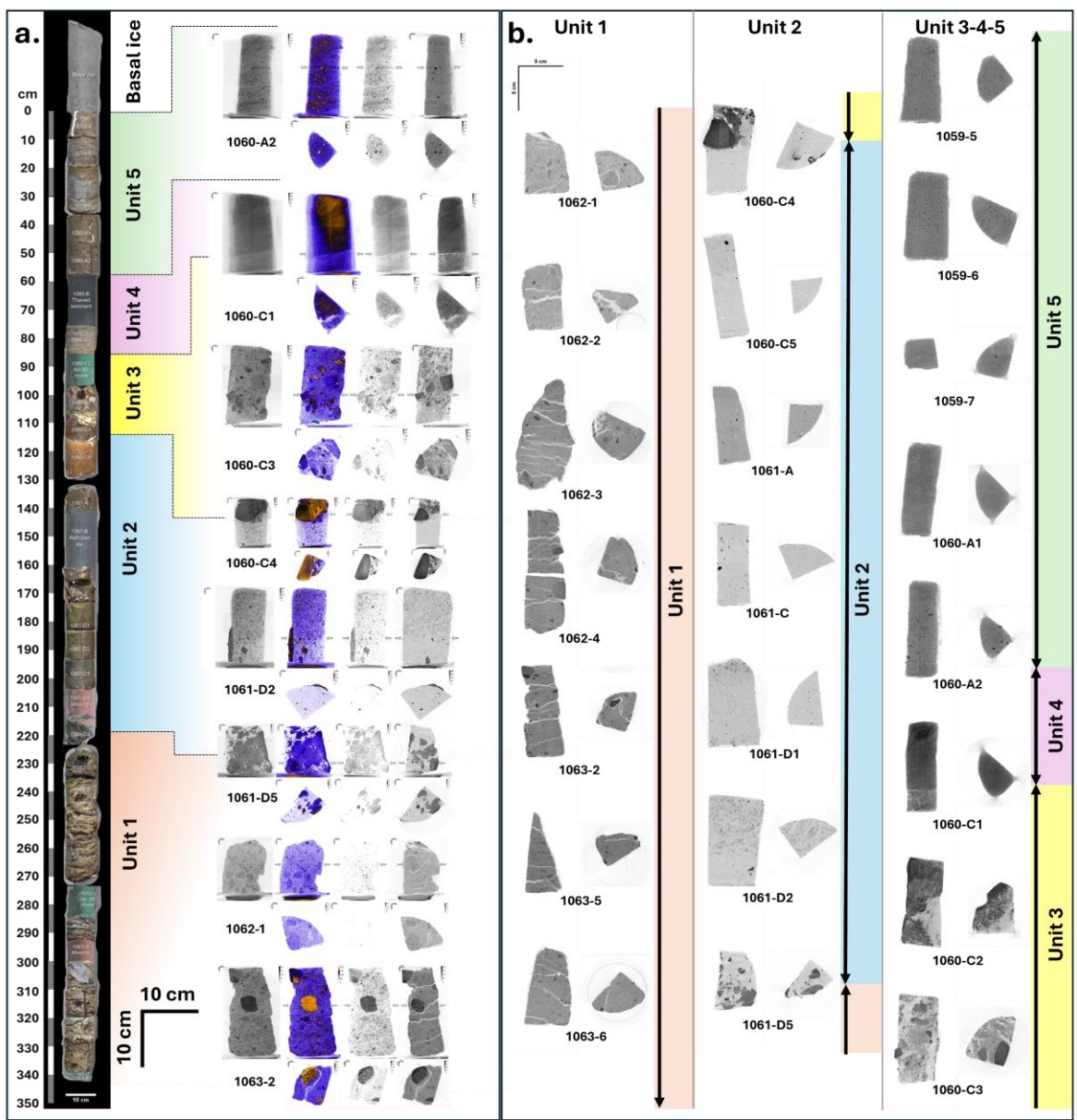

Figure 2. (a) Visualizations of selected CT scans representative of the five units described in Bierman et al. (2024). The 3D photogrammetric models are extracted from Bierman et al (2024). For each selected CT-scan we represent, in this order, a gray scale 3D rendering, a false color 3D rendering, a half histogram grayscale filtered 3D rendering and a selected z-slice. Below these representations, selected xy slices corresponding to the green arrows are represented. All representations in a) are scaled similarly. (b) Z-slices and xy slices of all samples collected. For each sample the z-slice was taken exactly halfway through the scan and the xy slice exactly mid-height of the sample. In a) the selected samples represent both samples representative of the unit (1063-2, 1062-1 for Unit 1, 1061-D2 for Unit 2, 1060-C3 for Unit 3, 1060-A2 for Unit 5) and samples capturing transitions between units (1061-D5 between units 1 and 2, 1060-C4 between Units 2 and 3, 1060-C1 between Units 3 and 4). Note that 1060-C1 is both characteristic of Unit 4 and highlights the transition with Unit 3.

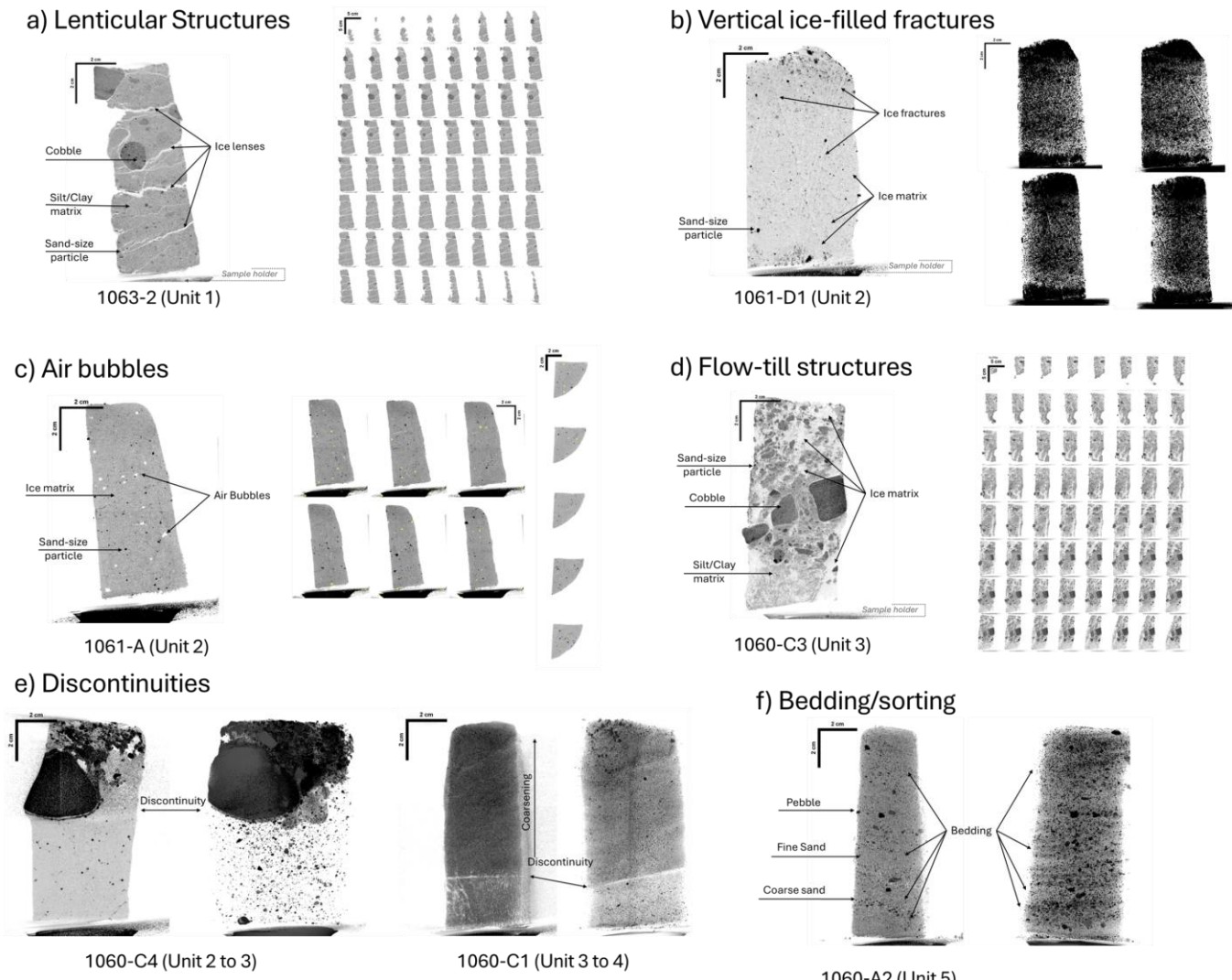

**Figure 3. Examples of characteristic features as observed by CT scan. a) Example of sub-horizontal ice lenses in Unit 1, poorly sorted sediment. These lenses connect in a braided lenticular pattern in all Unit 1 samples. The ubiquity of these ice lenses is highlighted on the right montage as the lenses penetrate the entire sample. b) Example of vertical ice-filled fracture in Unit 2. The right side of the image represents 4 different rotated views within a contrast-enhanced 3D reconstruction of the scan. Each view shows sub-vertical cracks at various angles within a matrix of sediment rich ice. c) Evidence of preserved air bubbles in Unit 2. The right side of the image represents 6 vertical slices and 5 horizontal slices through the sample with the air bubbles colored yellow for visualization. The slices used in visualization are regularly spaced through the sample. d) Characteristics of Unit 3 sediments show a general lack of horizontal structure and poor sorting within an ice-rice layer. The apparent chaotic nature of the material in Unit 3 is suspected to be the result of a flow-till event. e) Evidence of discontinuity between Units 2 and 3 (left) and 3 and 4 (right). The 2/3 discontinuity is marked by a change in ice content and sediment characteristics. The 3/4 discontinuity is marked by a change in layering and upward coarsening above the contact. f) Evidence of sorted beds in Unit 5. Beds appear to be principally characterized by alternating beds of well sorted fine and coarse sand.**

MicroCT scans show that the lowermost 7 samples (1063-5 to 1061-D5) define a homogeneous unit (Unit 1; 327-223 cm depth below the ice-sediment interface), previously defined as a diamicton (Bierman et al., 2024; Christ et al., 2021). CT analysis

reveals a variety of cryostructures (Figure 2a). These 7 samples are characterized by variably sized clasts (ranging from angular to sub-rounded and spherical to elongated, from sand to cobble size) in a finer matrix without bedding. Throughout the unit, sub-horizontal ice lenses cut through the sediment in a braided lenticular pattern (see example on Figure 2b and 3a). At the top of the diamicton, the CT scan of sample 1061-D5 (223-215 cm) captures a transition in the style of deposition. The upper part of the sample has a high ice content compared to the lower part of the sample which is sediment-rich like the rest of Unit 1. The transition is not discrete as the volume of ice increases upward within the fine-grained matrix material.

The next six and a half samples (1061-D2 to the lowest portion of 1060-C4) comprise Unit 2 (215-108.5 cm). They are characterized by low density (<1.2 g/cm$^3$) and high ice content (>80%, method described in Bierman et. al 2024) with sediment interspersed throughout. The µCT scans show that sediment is generally fine-grained with few clasts and faint bedding tilted at approximately a 45° angle from horizontal. The tilted bedding is more apparent in some samples (1060-D2, 1061-D1, and 1060-C4). Vertical fractures filled with clear ice create intersecting planes in the sediment-laden ice (e.g., 1060-D1, Figure 3b). Some samples also contain visible air bubbles (1061-C2; 1061-A, Figure 3c; 1060-C5) (Figure 3c). The contact with the unit above (1060-C4) is a discontinuity marked by a sudden change in ice content in the form (Figure 3e).

Unit 3 (108.5-88.5 cm) is not stratified. The tops of sample 1060-C4 and sample 1060-C3 contain pebble-sized clasts in a silt-sized matrix with lower ice content (23%) than other samples in Unit 3 (Bierman et al., 2024). The presence of a large (3-4 cm in diameter) clast in 1060-C4 marks the transition from Unit 2 (Figures 2 and 3d). As seen on the z-slice of the CT scan, the contact between the clast and the ice-rich material below is characterized by a layer of cleaner ice, close to the clast and a silty layer further away. It is unclear though if that structure developed syndepositionally or after deposition. Clast angularity ranges from sub-angular to sub-rounded. From the bottom of this unit (1060-C4) toward its transition to unit 4 (1060-C1), the grain size appears to decrease. Sample 1060-C2 and the lower portion of 1060-C1 are characterized by the presence of a deformed fine-grained bedding with high ice content (49%) (Bierman et al., 2024). There is a very distinct unconformable contact in sample 1060-C1 (at an angle of 11°) with the upper sediments, distinguishable by µCT scans (Figure 2 and Figure 3e). Directly below the contact, bedding is nearly vertical which continues into 1060-C2 (Figure 2b).

Both units above Unit 3 are similar in structure, containing well-sorted bedded fine sand (Figure 2 and Figure 3f). Unit 4 (88.5-55.5 cm) includes sample 1060-C1 (the part above the contact) and 1060-B (not scanned). Bedding in these samples is distinct, parallel, and sub-horizontal with a ~15° dip. The well-sorted fine-grained nature of the deposit defines this unit. Samples 1060-A2 to 1059-5 define Unit 5 (55.5-0 cm). These samples coarsen upward into gravelly sand from the fine-grained sands of Unit 4. Bedding is well defined in the lower samples in this unit (notably 1060-A2, Figure 3f) and becomes harder to distinguish in the top-most sample scanned (1059-5). The layers also dip at 15°. There are some small pebble-sized clasts, which range from sub-rounded to sub-angular, that form individual beds. Well-sorted sand-sized grains that coarsen upward with well-formed bedding define the upper most part of the core and the top of unit 5 (Figure 3).

 **4.2 X-ray Diffraction**

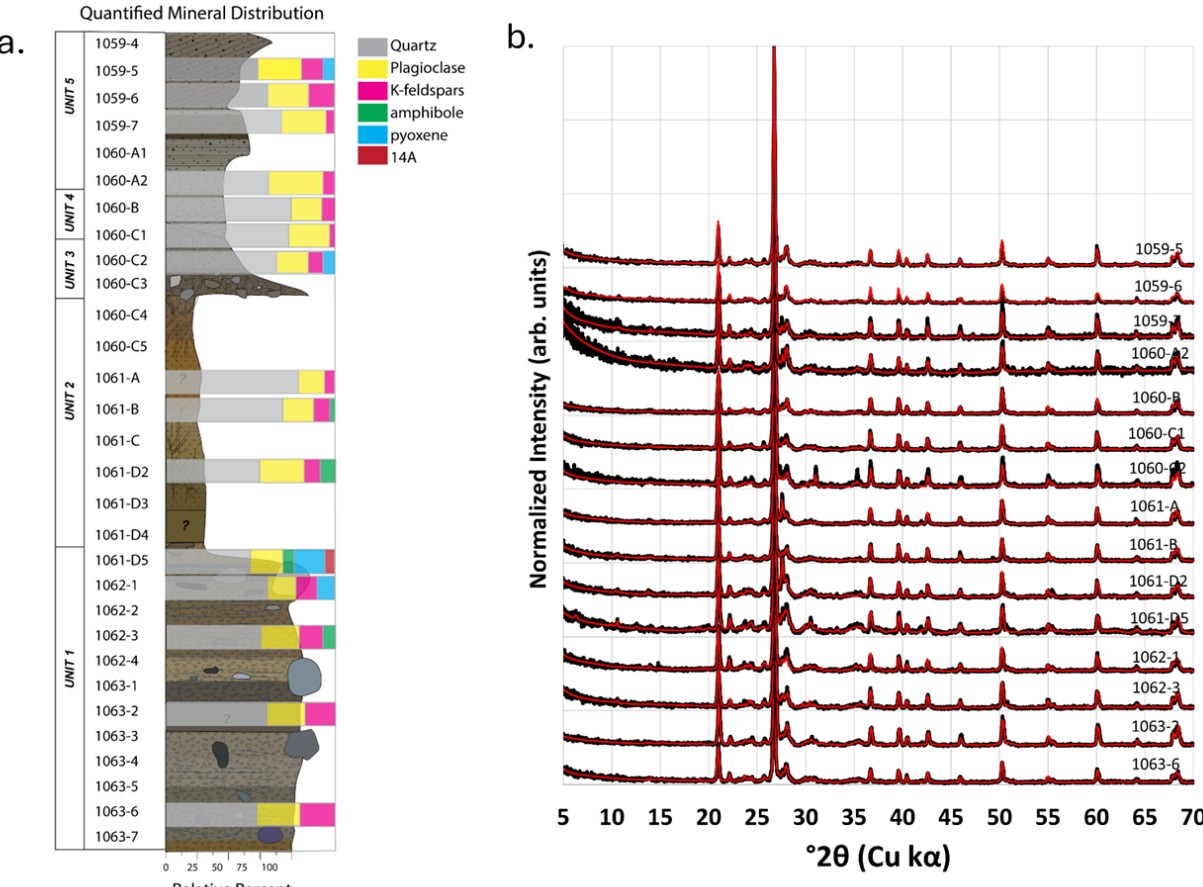

**Figure 4. (a) Relative mineral assemblage of 15 samples shown with corresponding color bars modeled using the Rietveld method (Rietveld, 1969). Stratigraphic column adapted from (Bierman et al., 2024). Results are given in Table 2. (b) Diffractograms of all 15 samples measured on the Rigaku MiniFlex II. Modeled diffractograms shown in red were calculated using the Rietveld algorithm included in PDXL-2: Integrated X-ray powder diffraction software. X-Ray intensities are normalized to the quartz peak and therefore represented in arbitrary units (arb. Units). "14A" corresponds to a crystalline phase with a dominant 14A reflection, similar to clinochlore.**

XRD analysis indicates little change in relative mineral abundance between units (Figure 4, Table 2). Throughout the core, the mineralogy is dominated by quartz with lesser amounts of plagioclase and potassium feldspar (Table 2). The average quartz relative abundance is 63.6%±3.9% with a minimum of 49.6% ± 0.7% (1061-D5, Unit 1) and a maximum of 80.6% ± 1.1% (1061-A, Unit 2). We found plagioclase in all units with an average of 22.4% ± 3.5%, a minimum of 14.8% ± 1.1% (1061-A, Unit 2), and a maximum of 31.8% ± 1.3% (1060-A2, Unit 5). Potassium feldspar was detected in all samples except 1061-D5 (Unit 1), the average was 10.5% ± 3.0% with a minimum of 3.1% ± 0.2% (1060-C1, Unit 4), and a maximum of 19.4% ± 0.8% (1063-6, Unit 1). The relative amount of quartz increases from the bottom to the top of Unit 2 although we only have analyses 3 samples. For all other units, the relative proportions of quartz compared to other minerals remains similar.

**Table 2. XRD quantification (in %) of bulk sediment from representative samples. Mineralogy reported by mineral group**

| | | | | Mineral Content (wt. %) | | | | | |
|---|---|---|---|---|---|---|---|---|---|
| Sample | Unit | Quartz | Plagioclase[e] | K-feldspar[f] | Amphibole[g] | Pyroxene[h] | Clay[i] | Sum | red. $\chi^{2j}$ |
| 1059-5 | 5 | 62.2 (1.1)[k] | 23.1 (0.8) | 14.2 (1.2) | *n.d* [l] | 0.4 (0.2) | *n.d.* | 99.9 (3.3) | 3.5 |
| 1059-6 | 5 | 56.6 (1.7) | 25.3 (1.2) | 15.9 (1.7) | 2.2 (1.9) | *n.d.* | *n.d.* | 100.0 (6.5) | 4.8 |
| 1059-7 | 5 | 70.1 (0.9) | 26.0 (0.9) | 3.5 (0.2) | 0.5 (0.1) | *n.d.* | *n.d.* | 100.1 (2.1) | 4.4 |
| **Unit 5 Average** | | **63.0 (6.8)** | **24.8 (1.5)** | **11.2 (6.7)** | **1.0 (1.0)** | ***n.d.*** | ***n.d.*** | | |
| 1060-B | 4 | 75.0 (0.4) | 18.4 (0.4) | 6.6 (0.3) | *n.d.* | *n.d.* | *n.d.* | 100.0 (1.1) | 3.8 |
| 1060-C1 | 4 | 73.7 (0.6) | 23.2 (0.6) | 3.1 (0.2) | *n.d.* | *n.d.* | *n.d.* | 100.0 (1.4) | 3.8 |
| **Unit 4 Average** | | **74.4 (0.7)** | **20.8 (2.4)** | **4.9 (1.8)** | ***n.d.*** | ***n.d.*** | ***n.d.*** | | |
| 1060-C2 | 3 | 66.1 (1.0) | 18.3 (1.0) | 9.8 (0.6) | *n.d.* | 5.7 (0.5) | *n.d.* | 99.9 (3.1) | 4.7 |
| 1061-A | 2 | 80.6 (1.1) | 14.8 (1.1) | 4.7 (0.3) | *n.d.* | *n.d.* | *n.d.* | 100.1 (2.5) | 3.5 |
| 1061-B | 2 | 68.6 (0.8) | 19.9 (0.7) | 8.9 (0.5) | 2.3 (0.6) | *n.d.* | *n.d.* | 99.9 (2.6) | 3.5 |
| 1061-D2 | 2 | 57.3 (1.3) | 26.1 (1.5) | 9.5 (0.5) | 7.0 (0.8) | *n.d.* | *n.d.* | 99.9 (4.1) | 3.9 |
| **Unit 2 Average** | | **68.9 (11.7)** | **20.3 (5.7)** | **5.8 (2.6)** | **3.1 (3.3)** | ***n.d.*** | ***n.d.*** | | |
| 1061-D5 | 1 | 49.6 (0.7) | 20.2 (0.6) | *n.d.* | 5.6 (0.5) | 19.6 (0.7) | 4.9 (0.4) | 99.9 (2.9) | 3.3 |
| 1062-1 | 1 | 61.3 (1.3) | 17.2 (0.6) | 12.0 (1.0) | *n.d.* | 9.5 (0.7) | *n.d.* | 99.9 (3.6) | 4.0 |
| 1062-3 | 1 | 57.3 (0.8) | 22.7 (0.7) | 13.9 (0.7) | 6.1 (0.5) | *n.d.* | *n.d.* | 100 (2.7) | 3.6 |
| 1063-2 | 1 | 60.2 (0.8) | 23.4 (0.8) | 16.4 (0.8) | *n.d.* | *n.d.* | *n.d.* | 100 (2.4) | 3.3 |
| 1063-6 | 1 | 54.3 (0.8) | 26.3 (0.8) | 19.4 (0.8) | *n.d.* | *n.d.* | *n.d.* | 100 (2.4) | 3.0 |
| **Unit 1 Average** | | **56.5 (4.7)** | **22.0 (3.4)** | **12.3 (3.2)** | **5.8 (7.1)** | **2.3 (0.4)** | **1.6 (2.2)** | | |
| 1059-4[m] | - | 58.6 | 33.3 | 4.6 | 3.5 | bql | 100 | | |
| 1063-7 | - | 61.8 | 33.8 | 2.0 | 2.4 | n.d. | 100 | | |

[e] Plagioclase includes all plagioclase detected (albite, oligoclase, andesine)

[f] K-feldspar includes all K-feldspar detected (orthoclase, sanidine)

[g] Amphibole includes all amphibole detected (cummingtonite, hornblende, ferrosilite)

[h] Pyroxene includes all pyroxene detected (pigeonite)

[i] Clay includes micas, kaolinite and chlorite-like phases

[j] red. $\chi^2$ represents the goodness of fit and correspond to $\chi^2 = [\Sigma_i (I_{obs} - I_{calc})_i^2 / \sigma^2(I_{obs})_i] / (n - p)$; with I the intensity, $\sigma(I_{obs})$ the estimated error of the measure (fixed to 10% of the counts), n the number of points used for simulation and p the number of parameters estimated during the fit.

[k] Error in parenthesis is provided by PDXL and represents the standard error for individual phases. For averages, the error is 1SD when n>2 or as deviation from the mean when n=2

[l] n.d. "not detected"

[m] samples analyzed in Christ et al. (2021). Note that plagioclase and K-feldspars were summed as feldspar, no error or goodness of fit was provided. bql stands for below quantification limit as clay minerals were detected in 1059-4 but not quantified.

### 4.3 Scanning Electron Microscopy Imaging and Energy Dispersive X-ray Micro-mapping

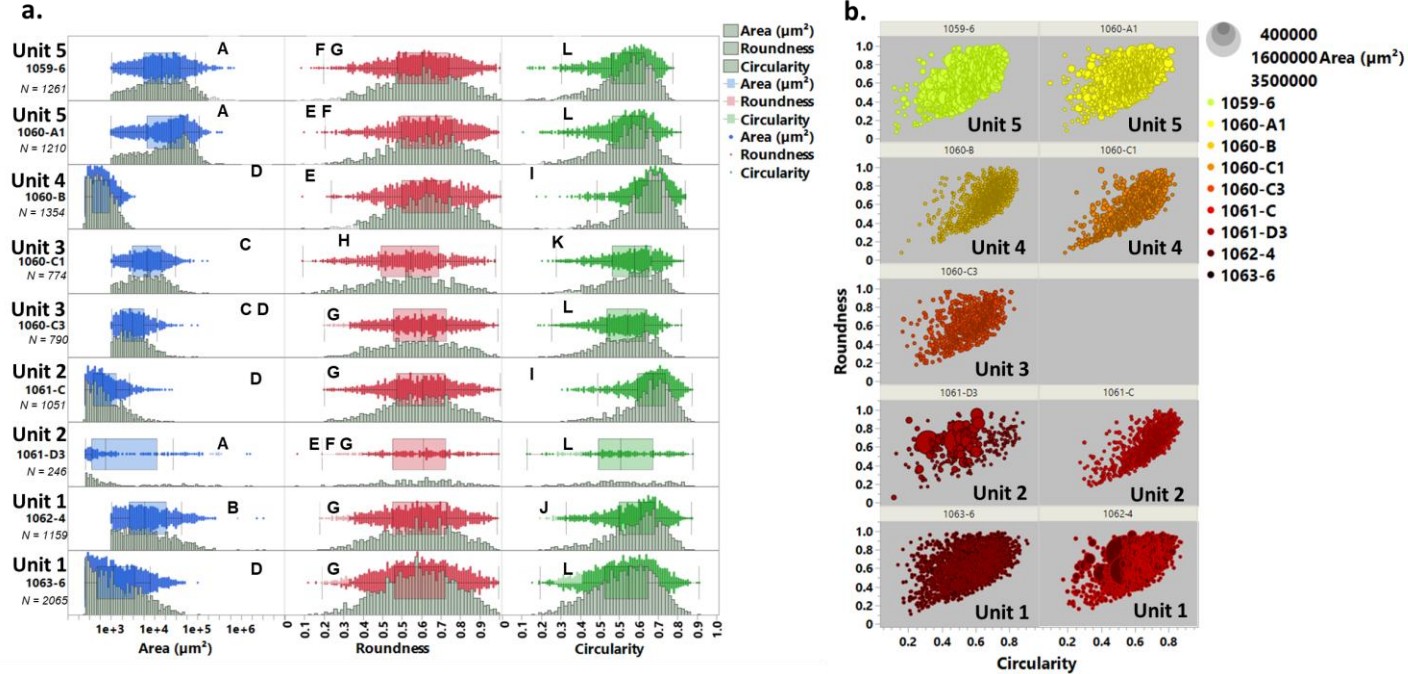

**Figure 5. Grain size and shape parameters extracted from low magnification SEM backscattered images of selected core samples. (a) Distribution of grain size (area), roundness, and circularity for individual samples, N is number of particles examined. For simplification, the area is represented in log scale. Roundness and circularity parameters were defined in section 3.4. The letters refer to the pairwise significance of means (described in 3.4). Samples assigned the same letters do not show a significant difference, samples that do not share the same letter are significantly different. (b) Relationship between particle size, circularity, and roundness for selected individual samples. The size of each individual circle represents the relative area of individual grains.**

Fine-grained particles (section area <2500 µm$^2$) in plan-view (corresponding to a 60 µm circular equivalent diameter) are present in all samples, though the range of grain sizes differs per sample (Figure 5a). Samples 1060-B and 1061-C contain 99.3% to 90.3% of particles with an area less than 2500 µm$^2$, respectively. Samples 1059-6, 1060-C3, 1061-D3, and 1063-6 have on average 58% ± 64% fines (area <2500 µm$^2$) and 1060-A1, 1060-C1, and 1062-4 have an average of 19.5% ± 2.3% fines. Samples 1059-6, 1060-A1, and 1062-4 show a much wider distribution of grain sizes (Figure 5a). Pair-wise statistical analysis of grain size distribution finds three sample groups: (A) 1059-6, 1060-A1 and 1061-D3, (D) 1060-B, 1061-C and 1063-6, and (B) 1062-4 as different from each other by particle size.

Roundness has a mean value of 0.62 ± 0.01 with a maximum mean value of 0.65±0.01 (1060-B) and minimum mean value of 0.58±0.01 (1060-C1), indicative of moderate roundness. While all distributions of roundness are symmetric and unimodal, a Tukey-Kramer mean comparison tests show that the grains in sample 1060-C1 (H) are significantly different from those in other samples and that the grains in sample 1060-B are also significantly different from those in 6 of the 9 samples (Figure 5).

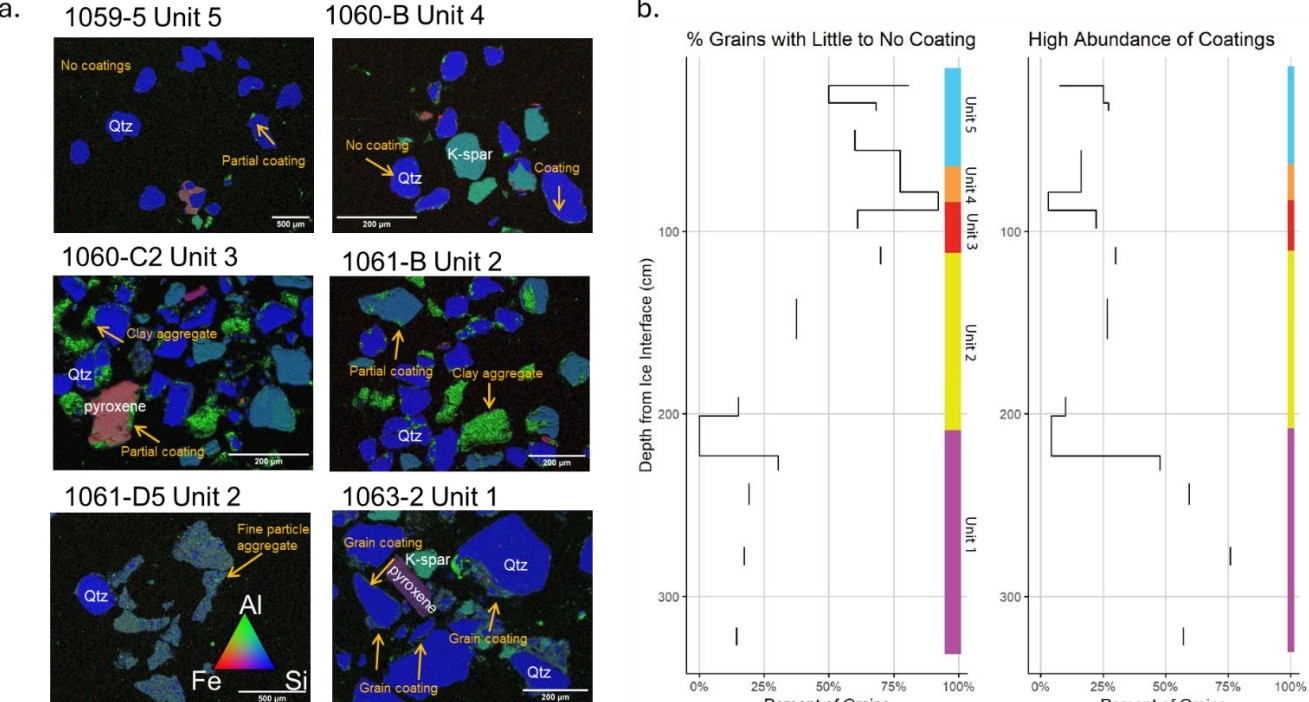

**Figure 6. (A) Scanning electron microscopy images showing mineral type and observed coatings. Blue indicates high silicon content (quartz); green is associated with aluminum (clays and feldspars), and red is associated with iron. (B) Abundance of grain coatings down core showing percentage of grains with little to no coating and percentage of grains with a high abundance of coatings (dominated and moderate categories combined). The colored bar shows unit assignments with depth.**

Circularity is consistently left-skewed and biased towards higher circularity values; all samples fail the Anderson-Darling test for normality. A Tukey-Kramer test splits the data into four distinct groups: (I) 1060-B and 1061-C, (J) 1062-4, (K) 1060-C1, and (L) 1059-6, 1060-A1, 1060-C3, 1061-D3, and 1063-6. These groups separate similarly to the grain size distribution except for sample 1060-C3. Comparing size, roundness, and circularity, larger grains tend to have a lower circularity and higher roundness (based on a significant negative and positive Pearson coefficient, respectively) (Figure 5b).

Elemental mapping analysis of SEM images by EDS reveals strong patterns between grain coatings and core depth. Grain coatings and fine particle aggregates are most abundant in Unit 1 and are present in all Unit 1 samples (Figure 6a, b) where they are composed of either individual clay minerals (Al-rich) or fine polymineralic fine aggregates (Si- and Al-rich). Unit 2 has a mix of fine particle aggregates, grain coatings, and clay aggregates, but not all grains have coatings and others are partially coated (Figure 6a). In this unit, the coatings and aggregates are dominated by Al-rich, clay-like minerals. Unit 3 has clay aggregates and grains that are partially coated (Figure 6a). Units 4 and 5 have fewer grains with coatings than the other units.

Mapping Si, Fe, and Al allows us to identify quartz (high Si), feldspar and clay (high Al), and Ca-Fe-Mg rich pyroxene and amphibole (high Fe) (Figure 6a). Quartz and feldspar, including plagioclase, dominate all samples. There is a positive

correlation (correlation coefficient, 0.7) between the "high abundance" grain coating category (grains with more than 25% of grain coating around their perimeter) and core depth (Figure 6b). Tukey-Kramer comparison by "high abundance" and unit reveals that Unit 1 is different from all other units and that Unit 3 has similarities with Unit 1 and all other units.

Grain surfaces display a variety of micro-textures. High-resolution SEM morphometric imagery of individual grains reveals that most quartz and feldspar grains have a variety of shapes, with two main endmembers: fresh angular grains with high relief, and round and oblong grains. Some grains have intermediate shapes between these endmembers (Figure 7). Round and nearly round grains are rarely pristine and display superimposition of different textures. We find that a considerable number of grains display high relief features including deep troughs, sharp angles, conchoidal fractures, and adhering particles (Figure 7 B: 1059-4, E: 1059-6, H: 1063-7). Some grains show V-shape percussion cracks (Figure 7 A: 1059-4, G: 1063-7) and a few grains display bulbous edges (Figure 7 D: 1059-4). Many grains display dissolution pits (Figure 7 F 1060-C3), multi-micron-thick coatings containing K and Fe (Figure 7 G: 1063-7, H: 1063-7), and quartz overgrowths (Figure 7 A: 1059-4, F: 1060-C3). We found that grains from most sedimentary units were homogeneous in terms of shape and texture, except for samples from Unit 1 that displayed more abundant weathering features than other units, in agreement with other SEM observations from polished grains (Figure 6).

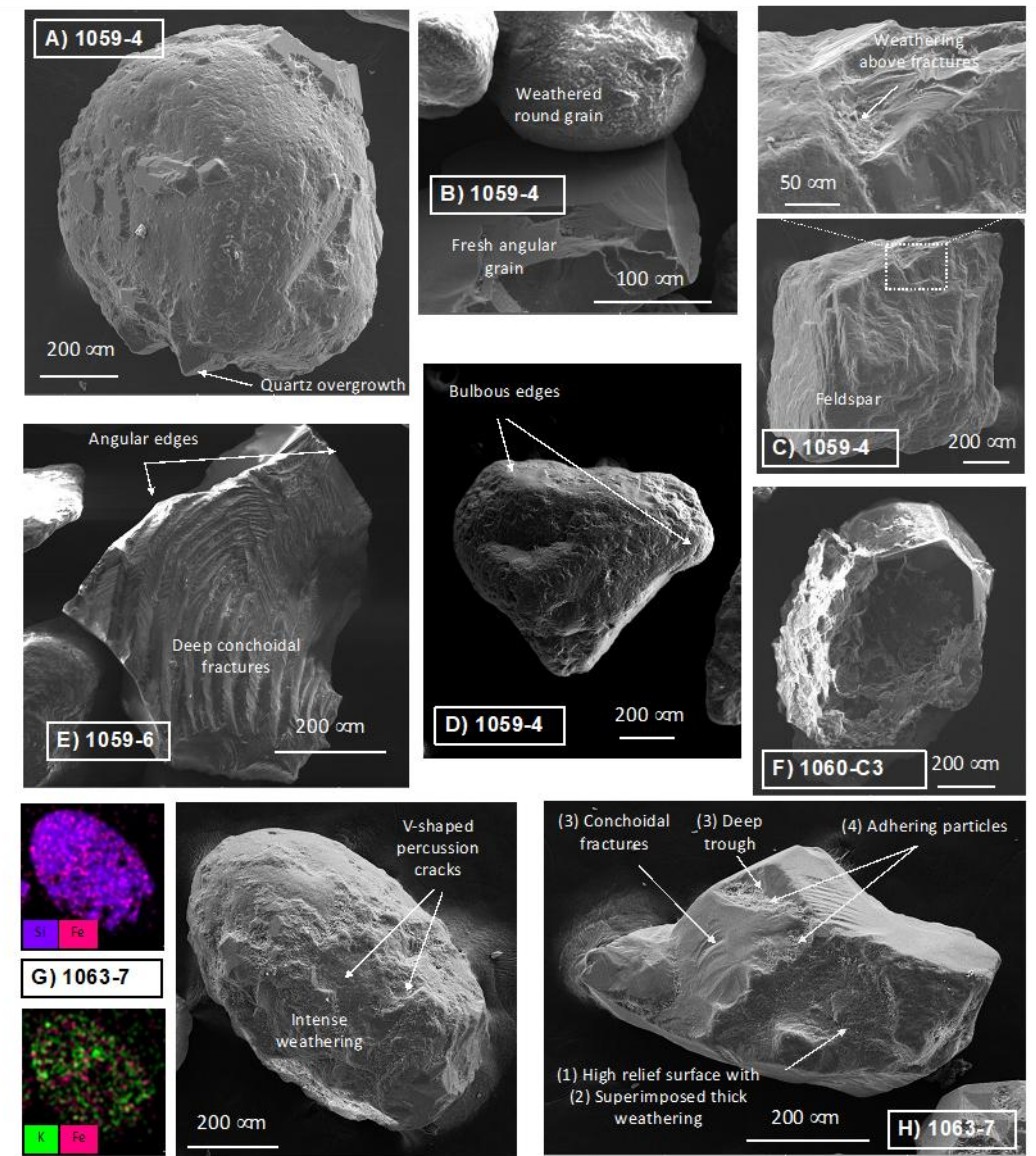

**Figure 7. SEM imagery from samples of Camp Century grains. A) 1059-4: round quartz grain with deep grooves, multiple v-shape percussion cracks and quartz overgrowth, the latter appears younger. Water transport followed by minor aeolian action. B) 1059-4: upper grain is rounded quartz with significant weathering and solution pits. Lower grain is a fresh angular quartz grain, with features consistent with glacial crushing and adhered fine-grained particles. C) 1059-4: K- feldspar angular grain with weathered surfaces. Top detail shows weathering features superimposed on pre-existing fresh fractures. D) 1059-4: Subangular quartz grain with multiple percussion cracks and bulbous edges. E) 1059-6: Angular quartz grain with two generations of conchoidal fractures- older on top left, top right and bottom; fresher in center. F) 1060-C3: subrounded quartz grain with weathering dissolution features and quartz overgrowths, G) 1063- 7: oblong quartz grain with remnant v-shaped cracks, significant weathering/coating features, and fresher surfaces. Backscattered electron analysis shows the presence of Fe and K in the thick micrometric coating. H) 1063-7: angular quartz grain with 1) an initial high relief surface affected by 2) a thick weathering rind, then later 3) deep trough and conchoidal fractures covered by 4) adhering particles.**

## 4.4 PCA and K-means Clustering

K-means clustering analysis shows optimal clustering occurs with 5 clusters (Figure 8a, b). There are similarities with this clustering and our unit assignments based on physical stratigraphic observations, most notably, Unit 1 samples are clustered tightly together. There is a mixing of Units 3, 4, and 5 samples across Clusters B and C. Unit 2 has samples within 3 different Clusters A, B, and D (Figure 8b).

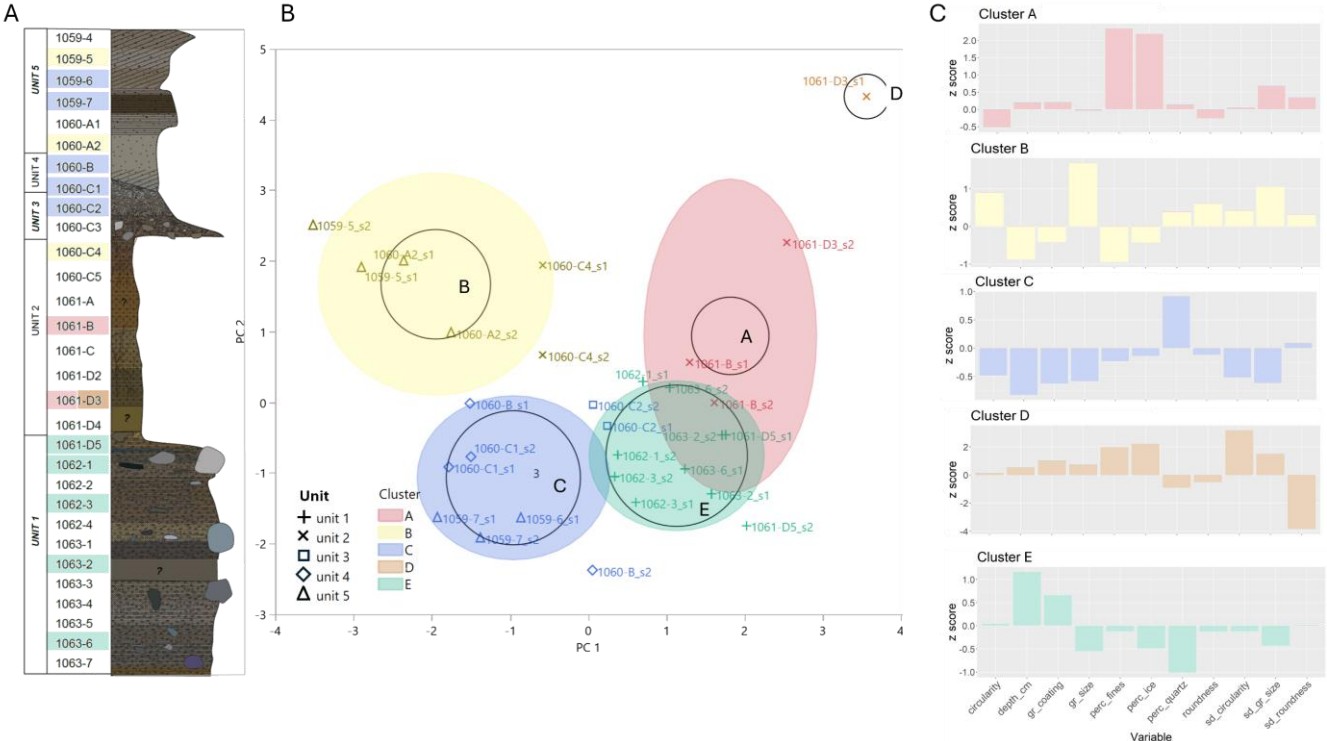

**Figure 8. (a) Stratigraphic column with samples highlighted according to the K-means clustering (b) K-means clustering analysis on multiscale physical, mineralogical and geochemical data. The data are plotted showing the first two principal components (PC 1 and 2), and are categorized into 5 clusters; clusters are color-coded and assigned a letter. Unit assignments are indicated by shape. (c) Z scores of each cluster are shown, which indicate how many standard deviations (y-axis) the mean of each variable in the cluster differs from the observed mean across all data. All z score graphs have the same x-axis variables in the same order as shown with**
**the Cluster E graph. The variables measured are: circularity, depth_cm (depth from ice interface in centimeters), gr_coating (nominal evaluation of presence of grain coating), gr_size (grain size by area), perc_fines (percent fine grained material), perc_quartz (percent quartz), roundness, sd_circularity (standard deviation of circularity measurement), sd_gr_size (standard deviation of grain size measurement), and sd_roundness (standard deviation of roundness measurement).**

Significant variables associated with PC1 include depth, percent quartz, grain coating, and percent fines. As depth is relevant
in stratigraphy and interpretations of stratigraphic deposits, we elected to retain depth as a variable to include as much information as possible in the algorithm. This principal component can be summarized as relating to transport processes, as percent fines and percent quartz often reflect sorting that occurs during transport and grain coatings are stripped in part or entirely during transport processes. Significant variables associated with PC2 are grain size, standard deviation of grain size,

standard deviation of circularity, and circularity to a lesser extent. This component represents aspects of grain morphology.

According to PC1, transport process, there is a larger separation between Clusters A and B than there is between E and C (Figure 8b). Along PC2, clusters A and B are separated from Cluster C and E. Cluster D contains one sample belonging to Unit 2 (1061-D3) and it is positioned at extreme ends of both principal components showing there is a strong difference between this sample and the rest of the data. The other observation associated with sample 1061-D3 (as two images were analyzed for this sample) is assigned to Cluster A but is positioned at high ends of both axes, as is the same for Cluster D.

The $z$ scores help us interpret which variables are responsible for placing samples into a particular cluster and provide insight into why there may be disagreements between our unit assignments and the clusters (Figure 8b, c). Z scores for Cluster A, made up of Unit 2 samples, show the most significant factors are percent fines and percent ice with positive $z$ scores of 2.3 and 2.2 respectively (indicating the number of standard deviations that values for these variables are from the population mean in this cluster) (Figure 8c). This means Cluster A samples (Unit 2) have more fine-grained particles and more ice on average than

other samples. Cluster B samples (mainly Unit 5) have larger grain size with slightly more size variation and slightly higher values of circularity on average (grain size $z$ score =1.7, sd grain size $z$ score =1.1, and circularity $z$ score = 0.9). Samples in Cluster C (Units 3, 4, and 5) can be most succinctly described as having more quartz than samples in other clusters (percent quartz $z$ score = 0.9). Cluster D (1061-D3, Unit 2) is quite different from the other clusters as it has the largest $z$ scores. This sample has low variability in rounding and high variability in circularity with high ice content (sd roundness $z$ score = -3.8, sd

circularity $z$ score = 3.2, and percent ice $z$ score = 2.2). Lastly Cluster E (Unit 1) $z$ scores show these samples are at a greater depth and have less quartz on average (depth $z$ score = 1.2, percent quartz $z$ score = -1).

## 5 Discussion

The geochemical and physical analyses we have made of the sub-ice material indicate that cores taken from the bed of ice sheets have the potential to provide unique insight into past conditions extending deeper in time than records preserved in the

glacial ice above. As Earth's average temperature rises quickly, such data are increasingly important because they are useful for understanding how ice sheets behaved in the past as climate changed (Gemery & López-Quirós, 2024).

### 5.1 Subglacial Core Stratigraphy: Synthesis of Physical, Chemical, and Mineralogic Observations

The combination of geochemical, mineralogical, and physical data collected from the Camp Century subglacial materials across different spatial scales characterizes in detail 5 distinct stratigraphic units building on and refining the prior work of

others (Bierman et al., 2024; Christ et al., 2021, 2023; Harwood, 1986; Whalley & Langway, 1980). From these data, we infer that different surface processes deposited these materials during times when the land beneath Camp Century was glaciated and when it was not.

Our data demonstrate that after deposition, Unit 1 was affected by weathering processes near Earth's surface. The unit has abundant grain coatings (Figure 6b) formed by percolating liquid water during at least one interglacial period (Marschalek et

al., 2024). Warm conditions facilitated the accumulation of grain coatings on the sediment grains before subsequent cooling

occurred burying Unit 1 beneath ice. The formation of permafrost halted the formation of grain coatings but preserved those already present in the sediment. The sub-horizontal, braided, lenticular cryostructures, consistent with syngenetic permafrost formation (French & Shur, 2010), suggest little influence of liquid water in Unit 1 after permafrost formation. However, without dating or contextual evidence we cannot rule out that they developed in a subglacial setting or that cracking could

have been related to stresses induced by overlying ice.

Unit 2, between Units 1 and 3, marks a stark transition from the sediment-dominated till to an ice matrix with dispersed sediment. The sediment content of Unit 2 and the horizontal alignment of grains seen in samples 1061-D2, 1061-D1 and 1060-C are typical of BIL (Knight, 1997). The source of the sediment interspersed in the ice is likely the till below (Unit 1) because they share similar mineralogy (presence of pyroxene) and fine particle aggregates, possibly glacial flour (Gresina et al., 2023;

Knight, 1997). Other possible origins of Unit 2 include interglacial firn or basal ice from a later ice advance, both allow for deep weathering of Unit 1 (Hambreky et al., 1999).

The transition between Unit 2 and 3 occurs as a discontinuity marked by the presence of a cobble (ca. 3-4 cm in diameter) and pebbles. The position of both suggests gravitational deposition on top of a "soft" ice layer as the cobble is lower than the pebble material. From the bottom of Unit 3 (1060-C4) toward its transition to Unit 4 (1060-C1), grain size decreases and no

clear ice structure (layering or lenses) is visible. These observations imply that the deposition of Unit 3 occurred via mass movement in the presence of liquid water, depositing diamicton above Unit 2 before both froze (Brevik & Reid, 2000). The mechanism causing mass movement is unknown, it could be due to undercutting from a small stream or erosion of permafrost polygons.

The normal grading of Unit 3 likely resulted from sorting as the sediments flowed down gradient, which has been noted by

others in flow till deposits (Brevik & Reid, 2000). The top of Unit 3 has distinct layering and an absence of large clasts which distinguishes it from the lower section of the unit. The layering in sample 1060-C2 and under the contact in 1060-C1 could be a marker of multiple flows or sorting within one flow (Figure 2b). The lack of coarse clasts in the upper part of Unit 3 may reflect limited flow capacity capable of moving only smaller particles during these later flow events or of a change in source (Brevik & Reid, 2000). The paucity of grain coatings in Unit 3 compared to Unit 1 (Figure 6), could indicate that mass

movement disrupted grain coatings or mixed sediment from Unit 1 with sediment from the upper Units, 4 and 5. In the absence of grain orientation and with the limited lateral context and size of the cores, other genetic origins for Unit 3 cannot be discarded, such as partial melt and reworking of nearby till by stream currents or the collapse of a subglacial stream bank. The presence of organic remains in all samples (Christ et al. 2023) indicates that at least some material in Unit 3 was sourced subaerially.

Unit 1, till, and Unit 3, a mass movement deposit, have similar mineralogy and some similar characteristics in the μCT scans (Figure 2). These similarities are seen in sample 1060-C3, the bottom of Unit 3, which is characterized by variably sized clasts in a fine-grained matrix with no evidence of bedding. The higher ice content and the difference in cryostructures between Units 1 and 3 supports the hypothesis that Unit 3 was deposited in or by liquid water (Larson et al., 2016; Menzies J., and Reitner, J. M., 2016). The presence of Ca-Fe-Mg pyroxene in both units 1 and 3 suggests that Unit 3 was derived from material

compositionally similar to Unit 1. Pyroxenes readily weather from sediments (Goldich, 1938) so their inclusion in a sediment with high quartz content could reflect the source rock composition, limited weathering, or mixing with localized sources of different mineralogy. Overall, similar sedimentary structures and mineralogy suggest that Unit 3 was originally a part of the subglacial till (Unit 1) that was subject to mass movement resulting from saturation by liquid water during interglacial conditions. Micro-texture analysis (Figure 7) from Unit 1 suggests the same succession of events deduced from meso and

macro-observations: initial glaciation and till deposition, then ice-free conditions allowing weathering and the development of grain coatings.

In Units 4 and 5, sorting, minimal grain coatings, and the presence of bedding are indicative of fluvial sediment transport sufficient to abrade grain coatings and deposition from moving water (Figure 2). Multiple lines of evidence suggest that sediment in Units 4 and 5 was derived from the erosion of Unit 1 (glacial till) and transported by moving water. Grain textures

shown in SEM imagery (Figure 7) show that Units 4 and 5 contain many grains with fresh surfaces, glacial crushing features, and sharp angles. This suggests that they originate from a proximal source of glacial sediment, and were not transported over a long distance, probably less than a few km (Mahaney, 2002). The fluvial system initially transported very fine-grained sand (Unit 4) followed by larger grains representing an increase in system energy or a different sediment source (Unit 5). Grain morphology in these units is a mix of round and angular grains, with minimal change from Units 4 to 5 (Figure 5).

Most of the more rounded grains in Units 4 and 5 display significant weathering features and glacial crushing, implying that their oblong and round shapes were probably inherited from previous ice-free cycles (pre-Pleistocene sandstone deposits are observed in northern Greenland (Gregersen et al., 2022). Hence, fluvial transport may not be the only cause of grain rounding (Kuenen, 1959). Unique to Units 4 and 5 are grains with coatings only in cracks or concave structures that protected the coatings from abrasion during transport, preserving evidence of past weathering (Figure 6). This implies that these sediments

originally had more substantial grain coatings that were stripped during fluvial transport – consistent with geochronologic data showing that minerals in Unit 1 and Unit 5 have similar cooling ages (Christ et al., 2023).

The lack of the heavy minerals in the fluvial sediments could reflect hydraulic sorting, with the result that denser minerals, including pyroxene and amphibole, were deposited elsewhere, perhaps with larger grain size fractions (Malusà et al., 2016). This would explain the lack of heavy minerals in the finer grain size fraction (Unit 4) and the presence of pyroxene at the top

of Unit 5 where coarser grain sizes are more common (Garzanti, 2017; Malusà et al., 2016).

Statistical analysis (PCA and K-means, Figure 8) clearly separates materials based on glacial and non-glacial origin. PC1, relating to transport processes, separates the data between glacially influenced Units 1 and 2 and fluvially driven processes characteristic of Units 3, 4, and 5. Unit 3 samples overlap with Cluster E (Unit 1) which supports our interpretation that these two units both originate from the same till. PC2, relating to grain morphology, does not show any distinct trends relating to

our unit assignments (Figure 8). The grain morphology analysis shows a similar complexity (Figure 5) which indicates intra-unit variability not seen on the meso or macro-scale.

Unit 2 is the most scattered across the PCA which suggests that it has a different and more complex history and/or formation process. The SEM data for Unit 2 echoes this complexity as it has the most intra-unit variability. Despite the variation within

the unit, some samples closely resemble other units at the micro-scale. Sample 1060-C4, at the top of Unit 2, is most similar
to samples from Unit 5 as it has minimal grain coatings and above-average grain size (Figure 6c). Conversely, sample 1061-
B, near the middle of Unit 2, has aluminum-rich clay coatings which are also present in samples from Unit 3. On the PCA,
1061-B samples are situated near Unit 1 and 3 samples, corroborating the similarities seen at the micro-scale. Samples closer
to the contact with Unit 1 (1061-D3) are characterized by small particle aggregates that are unique to this section of core. The
1061-D3 samples plot at the high end of PC 1 and 2 emphasizing their distinction from other parts of the core (Figure 8b).

The spread of data from Unit 2 could be an indication of environmental changes represented by this ice unit or of mixing
between units above and below (Units 1 and 3). For example, we have characterized Units 5 and 3 as having quite different
transport mechanisms and diagnostic characteristics of both these units are present in Unit 2. The inclusion of particles with
characteristics like those in other units could imply that the changes seen throughout Unit 2 indicate variability in the source
of particles and the mechanism of particle inclusion in the ice.

**5.2 Proposed Sequence of Environmental Changes at Camp Century**

Camp Century subglacial materials preserve a sedimentary record of changing surface and subglacial processes. From our
multi-scale investigation of these materials, we present a sequence of ice retreat and advance events consistent with evidence
including documented surface exposure during MIS 11 (Figure 9) (Christ et al., 2023; Woznick, 2024) and abundant remains
of vegetation across the entire core (Christ et al 2021, Mastro 2023).

1.  *Initial conditions*: The bottom part of the core (Unit 1) contains sediment transported and deposited by glacial ice.
             Glaciation in northwestern Greenland abraded materials beneath the ice creating the basal till including cobble-sized
             clasts and fine rock flour.
         2.  *Retreat*: Ice retreated exposing the former bed of the ice sheet to surface processes (Christ et al., 2021, 2023). The till
             weathered and grain coatings formed (Christ et al., 2021, 2023). Sometime after the till weathered, Unit 2 was
deposited. Unit 2 may be preserved snow/firn from an interglacial or remanent basal ice from a later glaciation.
         3.  *Water and vegetation*: During interglacial conditions, the permafrost landscape was subject to freeze-thaw cycles as
             shown by the vertical lenses of clear ice in Unit 2. Till, saturated by water, flowed downslope and buried the ice
             forming Unit 3. Interglacial conditions supported plant growth (Christ et al., 2023) and the development of a fluvial
             system that eroded the upper portion of the flow-till deposit stripping grain coatings but not changing grain shape.
The fluvial system then deposited bedded sand, initially fine-grained and then coarser-grained material.
         4.  *Readvance*: When the climate cooled after MIS 11, ice covered the Camp Century core site (Christ et al., 2023). This
             ice is currently cold-based and non-erosive.

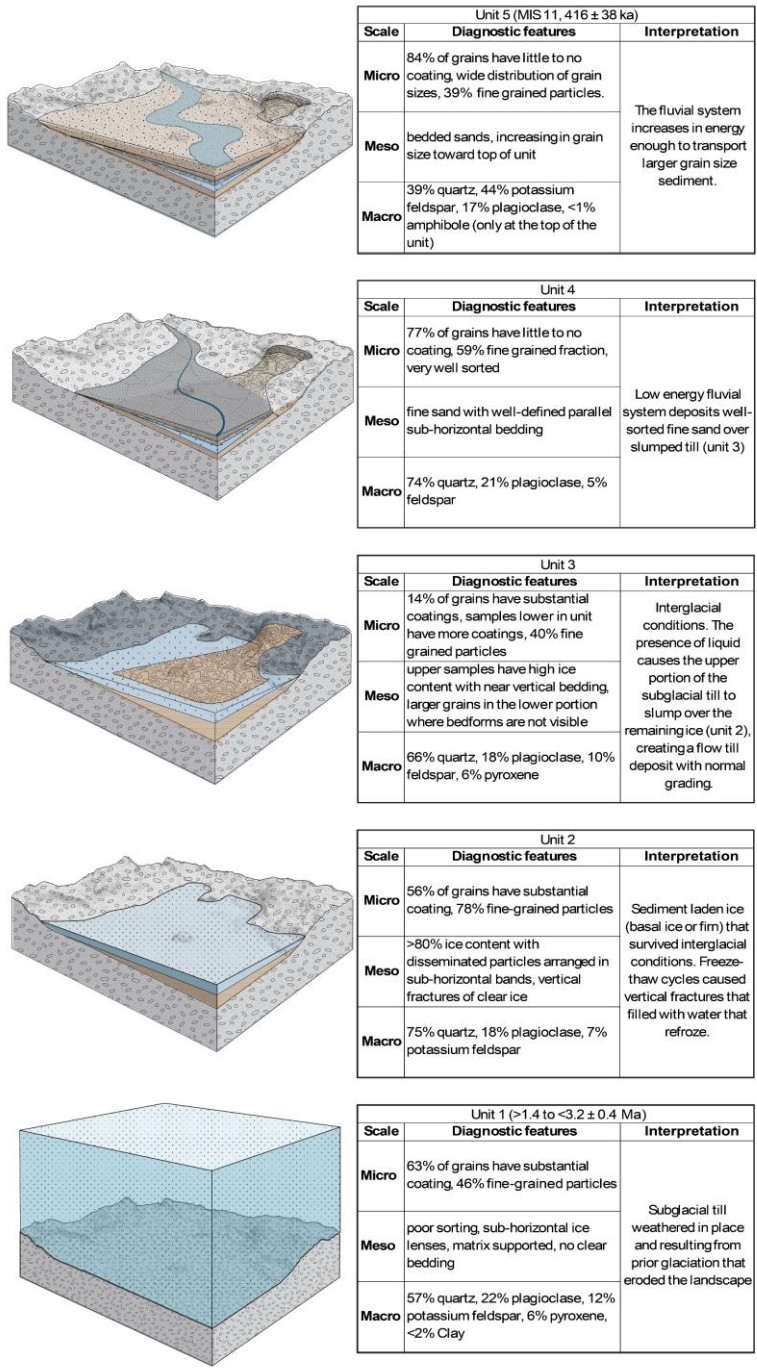

| Unit 5 (MIS 11, 416 ± 38 ka) | | |
|---|---|---|
| Scale | Diagnostic features | Interpretation |
| Micro | 84% of grains have little to no coating, wide distribution of grain sizes, 39% fine grained particles. | The fluvial system increases in energy enough to transport larger grain size sediment. |
| Meso | bedded sands, increasing in grain size toward top of unit | |
| Macro | 39% quartz, 44% potassium feldspar, 17% plagioclase, <1% amphibole (only at the top of the unit) | |

| Unit 4 | | |
|---|---|---|
| Scale | Diagnostic features | Interpretation |
| Micro | 77% of grains have little to no coating, 59% fine grained fraction, very well sorted | Low energy fluvial system deposits well-sorted fine sand over slumped till (unit 3) |
| Meso | fine sand with well-defined parallel sub-horizontal bedding | |
| Macro | 74% quartz, 21% plagioclase, 5% feldspar | |

| Unit 3 | | |
|---|---|---|
| Scale | Diagnostic features | Interpretation |
| Micro | 14% of grains have substantial coatings, samples lower in unit have more coatings, 40% fine grained particles | Interglacial conditions. The presence of liquid causes the upper portion of the subglacial till to slump over the remaining ice (unit 2), creating a flow till deposit with normal grading. |
| Meso | upper samples have high ice content with near vertical bedding, larger grains in the lower portion where bedforms are not visible | |
| Macro | 66% quartz, 18% plagioclase, 10% feldspar, 6% pyroxene | |

| Unit 2 | | |
|---|---|---|
| Scale | Diagnostic features | Interpretation |
| Micro | 56% of grains have substantial coating, 78% fine-grained particles | Sediment laden ice (basal ice or firn) that survived interglacial conditions. Freeze-thaw cycles caused vertical fractures that filled with water that refroze. |
| Meso | >80% ice content with disseminated particles arranged in sub-horizontal bands, vertical fractures of clear ice | |
| Macro | 75% quartz, 18% plagioclase, 7% potassium feldspar | |

| Unit 1 (>1.4 to <3.2 ± 0.4 Ma) | | |
|---|---|---|
| Scale | Diagnostic features | Interpretation |
| Micro | 63% of grains have substantial coating, 46% fine-grained particles | Subglacial till weathered in place and resulting from prior glaciation that eroded the landscape |
| Meso | poor sorting, sub-horizontal ice lenses, matrix supported, no clear bedding | |
| Macro | 57% quartz, 22% plagioclase, 12% potassium feldspar, 6% pyroxene, <2% Clay | |

**Figure 9. Proposed sequence of events that led to the accumulation of the 5 units. Diagnostic features summarized by scale of observation. Time constraints based on past work (Christ et al., 2021, 2023).**

### 5.3 Implications

Multiscale investigation of the Camp Century subglacial materials documents glacial and deglacial processes on the sub-ice surface of northwestern Greenland's over time. Systematic use of CT scanning enhanced our ability to describe the sequence of environmental change archived in the subglacial material. Such meso-scale observations allow us to identify changing environmental conditions, observe internal sedimentary and cryogenic structures, and provide an archive of 3D models of the samples that no longer exist in their original state. Macro-scale XRD mineralogic observations imply consistent sourcing of sedimentary material. Micro-scale data allow linkages of process-specific attributes to the meso-scale observations. Uncertainties remain regarding the explanation for intra-unit variability, the extent and duration of ice-free conditions, and the history and process of formation for Unit 2. Further investigation of the Camp Century sub-ice archive and future retrieval and analysis of subglacial materials have the potential refine the understanding of Greenland's paleoclimate and glacial history while further elucidating surface and weathering processes.

### Data Availability

Raw data from this project is archived in the Arctic Database and can be cited as: *Nicolas Perdrial, Paul Bierman, & Catherine Collins. (2024). 2019 - 2025 New Analyzes of the Camp Century Basal Sediment (Greenland): Mineralogy, Microscopy, Computer Tomography of the bottom 340cm of the core divided in 26 subsamples. Arctic Data Center. urn:uuid:2f6ee0ff-7444-4a76-85c4-909daac4312a.* (note that this is a preliminary citation.)

3D visualizations of the µCT scans are available for viewing and download in an online public repository:
 https://www.morphosource.org/concern/cultural_heritage_objects/000583438

### Funding

Funding provided by the United States National Science Foundation grant EAR-OPP-2114629 to Bierman and Perdrial and NSF-EAR-1735676 and 2300560 to Bierman and Corbett. Funding also provided by Geocenter-DK, grant GC 3-2019.

### Author Contributions

C.C., N.P., P.R.B., P-H.B., N.K.: Study design, data production, analysis, data interpretation, writing. C.C., N.P., P-H. B., N.K., P.C.K.: Grain analysis, scanning electron microscopy data acquisition. P-H. B., W.C.M., Y.M.: unpolished micro-scale

SEM data acquisition and interpretation. C.C., H.M., J.S., Sample handling. C.C.: Computed tomography scan acquisition. P.R.B, P.C.K, N.P., Funding acquisition.

**Competing Interests**

Paul R. Bierman is an editor on the special issue: Camp Century ice and sediment core: new science from a 1966 core that touched the base of the Greenland ice sheet (CP/TC inter-journal SI)

**Acknowledgements**

We thank Zoe Courville, CRREL, for help in CT scan acquisition, Jody Smith, Middlebury College, and Sebastian N. Malkki
(GEUS) for SEM and EDS assistance, Kristen Underwood, University of Vermont (UVM), for guidance with data processing and visualization, and MJ Moline (UVM) for assistance with creating CT scan visuals. We also acknowledge Andrew Christ for his work as part of the Camp Century team and early guidance on this project. We thank the editor of Climate of the Past and three reviewers for invaluable insights in improving this manuscript.

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
