# Peer review of "Characterization of the 1966 Camp Century Subglacial Core: A Multiscale Analysis"

_EGUsphere, 2024_

## Author Comment (AC1)

Collins et al. (2024) present a comprehensive multiscale analysis of the sub-glacial core collected from Camp Century, Greenland, analyzing physical, geochemical, and cryostratigraphic characteristics. The authors use X-ray diffraction (XRD), micro-computed tomography (μCT), and scanning electron microscopy (SEM), to identify and characterize five distinct sedimentary units, each with characteristic structures, mineralogical compositions, and cryostructures, and then use these data to infer a sequence of depositional environments, spanning glacial to interglacial to glacial. Their findings add to a growing body of work on this subglacial material and provide further context into Greenland's paleoclimate history and ice sheet dynamics, though some of their interpretations remain speculative without further supporting evidence.

The reviewer accurately captured the nature and aims of our work. We agree that our interpretations are limited to the evidence we have. Fortunately, new geochemical and biotic data are coming and we will include some of these (with citations to abstracts for now) so as to make interpretations more robust. Nevertheless, as described in the points below, we will carefully assess the level of speculation within our interpretation.

Overall, this paper is well-written and polished, and the data and generally analyses are robust. However, my primary critique of this work is that the authors' interpretations occasionally overreach given the data presented and some aspects of the presentation lack sufficient detail to clearly follow their logic. While the authors' interpretations are plausible, it appears they could represent one possible scenario among several.

This is a reasonable critique and we will work in revision to be more explicit about what data we have and what suggestions are more certain and which are more speculative.

This is particularly uncertain when inferring the timing of cryostructure formation or depositional environments from qualitative observations. I believe the paper provides a solid empirical foundation and presents a valuable dataset but lacks specific evidence for some interglacial interpretations.

We have significant additional data to support interglacial age assignments (see below) but no firm data on the age of cryostructures and will state that clearly.

I describe my main concerns with the interpretation of the sediment facies below and then follow up with minor concerns.

"The higher ice content and the difference in cryostructures between Units1 and 3 supports the hypothesis that Unit 3 was deposited in or by liquid water."
The specific differences in cryostructures are not clearly presented in this manuscript. The paper lacks plots or tables that detail ice content by depth, as well as quantitative descriptions of ice lens characteristics such as shape, spacing, orientation, and thickness.

Agree, we will do our best to provide more of these type of data in revision and reduce our emphasis on these structures.

These details could provide valuable clues to the origin of the ice lenses and would allow for comparisons with theoretical predictions. Including a figure (in the supplemental

material if necessary) that details and compares the various cryostructures within each unit in high resolution would greatly enhance clarity, as these structures are frequently referenced as evidence for interpreting the origins of each unit. Currently, it is challenging to contextualize the manuscript's claims alongside the physical evidence. The supplemental movies provide a helpful visual, but lack sufficient detail to quickly orient the reader.

This is an excellent suggestion we will do our best to incorporate specific details of cryostructures from the CT scans.

(As a note, I was unable to view the CT scan videos using an M1 Mac on either Safari or Firefox and ultimately had to download Chrome, which played them successfully.)

We deposited the scans in an NSF-supported repository (under review) and I have downloaded them in Safari on Intel, Firefox on PC and M2 Mac – not sure what the difficulty might be but not in our control.

**Regarding the Interpretation for Unit 2**
The authors suggest that Unit 2 is a ~1-m-thick remnant of basal ice (or even firn) preserved from a previous glaciation that survived interglacial conditions near the surface (e.g., Figure 8). This seems challenging to support without corroborating evidence, such as ice crystallography or age constraints.

We have neither ice crystallography nor any robust age constraints on the unit 2 but, initial luminesce data, neither complete or public suggests it is similar in age to the material above it (400-450 ky). The notion of firn/basal ice is supported by unpublished stable water isotope data. We will revise paper citing abstract with those data and change the wording to integrate possible alternative as this data is too broad to be integrated within this work and is currently the subject of articles in preparation

The resemblance to "basal ice" is unsurprising, given that this layer is located in subglacial frozen sediments near the ice-bed interface, akin to a frozen fringe. Theory predicts that a frozen fringe could grow meters thick and form ice lenses in basal till under freezing conditions (e.g., Meyer et al., 2023), following similar physics as frost heave in permafrost—an occurrence well-documented in the field (e.g., Christofferson and Tulacyzk, 2003; Fitzsimmons et al., 2024). A recent paper posits that clasts can migrate into ice through thermal regelation (Pierce et al., 2024), which could provide a mechanism to progressively incorporate till from the underlying layer. Ice crystal structure analysis could clarify whether this is highly sheared basal ice left over from a previous glaciation or ice grown through cryosuction, which would exhibit distinctive characteristics. Without such data, the timing of its emplacement remains speculative. One might imagine Unit 3 being deposited over Unit 1, with later ice lens growth. Based on the information provided, I don't see how it could be said with certainty.

The reviewer is correct – without direct geochronology we cannot for certain pinpoint the age of unit 2. Pore ice pH and conductivity of the core (Bierman et al 2024) show that they are dramatically different geochemically, we will make that clearer in the revision.

Additionally, while the presence of vertical cracks is intriguing and could potentially represent freeze-thaw cycles, I do not believe they are a smoking gun. This layer could have experienced any number of glaciotectonic events that induced brittle strain, and the history is difficult to deduce from a single core, which is essentially a point source. We agree that freeze-thaw is not the only possible explanation, we will expand this discussion.

**Regarding the Interpretation for Unit 3**
The authors describe Unit 3 as a slump deposit, partly based on the presence of normal grading. This interpretation is not readily apparent to me from the evidence provided in Figure 2. The core sections labeled as Unit 3 are divided between 1060-C3 and the lower portion of 1060-C1, with a substantial gap between them, which complicates a continuous interpretation. In the lower section (1060-C3), the sample is clast-rich, while the upper portion (1060-C1) contains mostly fines. This contrast presumably forms the basis for describing Unit 3 as normally graded. However, 1060-C3 does not visually appear graded; it lacks the clear stratification expected in a slump deposit and could simply represent basal till, with clasts suspended in a finer matrix.
We will re-examine evidence for normal grading and add to the revised ms additional data from other sources that also suggest this is a slump deposit. We will also revisit the terminology as we may have a different definition of slump deposits as the reviewer to consider the term debris flow as more appropriate. We however would like to point out that this portion of the core in in fact very complete and continuous as shown on figure 2 with the ct-scans of subcores 1060-C3, 1060-C2 and 1060-C1 in direct connection.

The missing sediment between 1060-C3 and 1060-C1 leaves some ambiguity in interpreting the contact between these layers. This discontinuity between a poorly sorted till and fines could indicate separate depositional events/mechanisms, weakening the argument for a continuous slump feature. Additional data—such as grain size distributions with depth or clast fabric/orientation relative to underlying till sections—would lend stronger support to the slump hypothesis (though it would still be difficult to ascertain from a single core without lateral context). Clast orientation specifically could help distinguish a traction till, characterized by horizontal shear, from a flow till, deposited downslope. In lieu of these constraints, alternative interpretations are plausible.
All of these ideas are wonderful but with a several inch wide core, not possible to test. We will work to provide viable alternative explanations while continuing to focus on the interpretations we consider best supported by our observations and data. We also want to point out that 1060-C2 is not missing and that the sample was analyzed by XRD, SEM and ctscan.

"The paucity of grain coatings in Unit 3 compared to Unit 1 (Fig. 5) could indicate that the slumping process disrupted grain coatings or mixed sediment from Unit 1 with sediment from the upper units, 4 and 5."

If Unit 3 was deposited as a basal till and subsequently covered by fluvial deposits, then the lack of grain coatings might be expected. My impression is that the main evidence for interpreting Unit 3's deposition as subaerial or subaqueous, rather than glacial, is the presence of weathered coatings in Unit 1 below, which suggests significant exposure at Earth's surface. This would imply that ice retreated, exposing Unit 1, before Unit 3 was subsequently deposited. An alternative hypothesis could be an ice readvance that deposited Unit 3. While the paleo-climatic sequence for this specific region isn't my specialty, are there regional constraints that would support this specific locale being ice-free for the full duration of the interval given by the age constraints of Units 1 and 5, or is this largely unknown?

We will clarify our description. We don't consider a re-advance as a likely source of a very thin (dm) layer of dimicton. Until our work, there was no understanding of the paleoclimatic sequence for Camp Century area because it's been buried by ice since 400 ky. Overall, we will better state and review what chronologic constraints exist for the sub-ice material.

"The sub-horizontal, braided, lenticular cryostructures are consistent with syngenetic permafrost formation, suggesting based on cryostratigraphy, little influence of liquid water in Unit 1 after permafrost formation (French & Shur, 2010)."

Based on the evidence provided in this manuscript, the timing of these lenticular cryostructures remains ambiguous. While the authors suggest they formed as syngenetic permafrost in a subaerial environment, it is equally plausible that they developed subglacially over a range of timescales. Without additional dating or contextual evidence, attributing them definitively to subaerial permafrost formation is speculative.

Agree, we will provide this alternative and make it clear there are other possible processes that could have formed

"Overall, the similar sedimentary structures and mineralogy suggest that Unit 3 was originally a part of the subglacial till formed below the ice (Unit 1) that was subject to slumping due to saturation of liquid water during interglacial conditions."

While I concur that this likely originated as subglacial till, the assertion that it is a slump deposit in an interglacial seems rather uncertain. As mentioned above, the timing here could have other possibilities. Even as a slump, perhaps it could be subglacial as well, perhaps representing the collapse of a canal's sidewalls.

We will be more specific in the logic we use to suggest the origin of unit 3 and cite other data. For example, luminescence ages on unit 3 show that it too was exposed at MIS 11 making a subglacial origin unlikely. WE appreciate the suggestion of a subglacial basal channel collapsing wall and will consider this option.

"Water and vegetation: During interglacial conditions, the permafrost landscape was subject to freeze-thaw cycles as shown by the vertical lenses of clear ice in Unit 2. Till, saturated by water, flowed downslope and buried the ice forming Unit 3. Interglacial conditions supported plant growth and the development of a headwaters fluvial system that eroded the upper portion of the flow-till deposit, stripping grain coatings but not

changing grain shape. The fluvial system then deposited bedded sand, initially fine-grained and then coarser-grained material."

The evidence presented here doesn't necessarily indicate a unique interglacial, subaerial depositional environment:

1. Ice lenses could have formed in these layers subglacially over an ambiguous range of timescales.
2. Unit 2 may have grown after the deposition of Unit 3. If Unit 3 represents a slump (though this is not convincingly demonstrated), it's plausible that it collapsed as a subglacial feature, potentially forming canal walls.
3. The introduction (Lines 60-62) states the presence of plant and invertebrate fossils in the core necessitates the site was ice-free during MIS11, but the authors do not explicitly state what units described in this paper contained signs of biological life. Fossils in Units 2-5 would significantly bolster the interglacial deposition argument, yet this isn't explicitly discussed in this manuscript. In particular, are there biological remains in Unit 3?

There is a lot to unpack here – some of which we can address with certainty, some we have other data that helps constrain the stratigraphy, and  the rest of which will, due to lack of data, remain uncertain. For example, we have found biological remains in every sample from the core – this is the focus of an upcoming paper. There are number of geochemical and isotopic measures showing the close similarity of units 1 and 3. But, unit 2 remains poorly understood with come suggestions from stable water isotopes that it could be firn deposited on the underlying unit 1 and then buried by unit 3. In revision, we will add the information about biologics requested by the reviewers and where appropriate discuss alternative hypotheses and where useful, present other data in support of our hypotheses.

**Line 172**: What evidence is there that this has undergone minimal deformation? Line 348 mentions deformed bedding, for instance, with no further explanation. Has previous work assessed the microstructure in detail? A citation would be warranted here.
This work is the first attempt at a partial microstructural analysis of the core using ct-scan. The evidence is visual. We will state this explicitly and add a reference to previous work when referring to the macrostructural analysis in this method section.

**Line 316**: "Our multiscale analysis supports, refines, updates, and provides more justification and detail regarding unit delineations proposed earlier (Bierman et al., 2024; Christ et al., 2021, 2023; Fountain et al., 1981)." It would be helpful to identify in the discussion how observations made in this paper agree or disagree with past interpretations of the units.
This is a good idea which we will implement in revisions.

**Line 322**: Technically, it is all subglacial material. Does this refer to Units 1 and 2? We will clarify that this refers to the entire subglacial core material

**Minor punctuation errors**

- **Line 47**: "1969)), but the sub-glacial material" → missing comma.
- **Line 172**: "minimally, if at all, deformed."

We will correct these errors.

Works cited

Meyer, C.R., Schoof, C. and Rempel, A.W., 2023. A thermomechanical model for frost heave and subglacial frozen fringe. *Journal of Fluid Mechanics*, *964*, p.A42.

Pierce, E., Overeem, I. and Jouvet, G., 2024. Modeling sediment fluxes from debris-rich basal ice layers. *Journal of Geophysical Research: Earth Surface*, *129*(10), p.e2024JF007665.

Christoffersen, P. and Tulaczyk, S., 2003. Response of subglacial sediments to basal freeze-on 1. Theory and comparison to observations from beneath the West Antarctic Ice Sheet. *Journal of Geophysical Research: Solid Earth*, *108*(B4).

Fitzsimons, S., Samyn, D. and Lorrain, R., 2024. Deformation, strength and tectonic evolution of basal ice in Taylor Glacier, Antarctica. *Journal of Geophysical Research: Earth Surface*, *129*(4), p.e2023JF007456.

---

## Author Comment (AC2)

The paper by Catherine Collins and co-authors presents a new rich dataset from the precious sediment samples collected at the base of the Camp Century ice core. The analyses include microCT, XRD, SEM-EDS and SEM imaging. The authors looked at the sedimentary facies in order to decipher the processes responsible for the deposition and preservation of sediments and infer some paleoenvironmental interpretation. The paper is well written and the figures have been made with great care, although several improvements can be done (see detailed comments below).

We appreciate reviewer's enthusiasm for our work.

This paper refers a lot to another one submitted to the journal *The Cryosphere* by Bierman *et al.* (still a preprint at the time this review is written).

This paper is now published.

Unfortunately, the reader needs to read the Bierman *et al.* paper to understand several things, such as the sample numbering and the stratigraphy. I think this essential information needs to be incorporated within this paper. For instance, the uppermost core section is the one on the right in the figure, which is not what one usually when presenting sediment cores. Incorporating panel B of figure 2 of the Bierman's paper would be very helpful.

With the editor's permission, we are happy to make this change. We were worried about replication but the reviewer makes a good point.

A disturbing feeling while reading the discussion section of this paper is that many arguments for the interpretations of the sediment facies are coming from previous papers. A clear distinction between the interpretations based on this new dataset, and the data presented elsewhere is needed.

This is a useful comment. We will work in revision to make clear what the new data set tells us that prior data did not.

My main concern is about the interpretation of the sedimentary facies. I'm not a specialist of glacial/proglacial/subglacial microfacies, but in such settings, it is critical to have an idea of position of the ice, the direction of the flows, the topography, the slopes, etc., to make a solid interpretation, which is impossible to obtain from a single sediment core.

We agree that our ability to make interpretations in limited because of the size of the core and our inability to place what we see in a broader landscape context. In particular, the core was never oriented and the sub-core stored without relative orientation. We will add a statement to the discussion making this clear.

Moreover, it is very likely that the sediments recovered at the bottom of 1350 m of ice experienced some glaciotectonism. This is not discussed in the manuscript, and it should, because it is not sure that the ice cap was cold based all the time, especially if one of the interpretations suggested by the authors is that a river flowed at this site. In short, I think that the observations remain interesting but that the facies interpretation is pushed too far.

It would be useful to have the input of a specialist of microsedimentological analyses of glaciogenic sediments.

We will consider glaciotectonism as we revise the manuscript. Others are conducting more detailed analyses of the CT scans which may provide more specific data based on clast orientation but that work is in progress and outside the scope of this particular paper. We believe the thermal state of the ice sheet and the presence of moving water when the area was deglaciated are two independent observations.

For the statistical interpretation, the choice of the variables considered is not well justified; for instance, why including the depth of the sample as a variable (it could be an explanatory data).

We may not fully understand the second part of the reviewer's question - "why it couldn't have been an explanatory variable?" We did use depth as one of our predictor (explanatory) variables. We will include justification for our variable choices.

Finally, there are many small mistakes and inaccuracies in the µCT dataset description making the paper arduous to read. There are details about the µCT data acquisition in the online public repository (congratulations for that effort), but it would be nice adding some information in the paper and/or in the repository in order to be able to reproduce the measurements (see my detailed comments below).

Unfortunately, I was unable to open all the movies made on processed images.

We will investigate with the repository why this might be the case.

While the movies are a very nice addition, it is quite impossible to analyse these images. The addition of images that can be handled by an image analysis software would have been better in my point of view, but this is already very nice.

Because of the nature of the core material, the detailed microstratigraphical analysis of the core was not achievable to date. It is a technical and methodological challenge that we currently investigating and consider worthy of a subsequent manuscript. The raw SEM and CT scan data is currently in review by the artic data center and should be accessible within a few weeks.

**Detailed comments:**

Line 65 : « sequence stratigraphy » has a specific meaning in sedimentology that is not related to the reality you want to describe. I suggest changing this by "the lithological succession".

Agree.

Line 166: "shrinking as sediment freezes". I'm not a specialist of these structures, but it does not seem logical to have shrunk when ice has a 10% higher volume. Can you expand on that?

Thank you for pointing this out. As written this sentence excludes part of what is included in French & Shur, 2010 which is explained as follows: "Reticulate cryostuctures are thought to reflect desiccation and shrinkage as sediment progressively freezes and moisture migrates to the advancing freezing front (e.g., Mackay, 1974)." We will add this citation and clarify the sentence to include this nuance.

Lines 1972-173: this statement is strong and should be supported by evidence or a reference to other studies.
We will add references and more supporting evidence.

Lines 1978-1982: these sentences belong to the introduction, not the method section.
We will move these sentences.

Method section: there is little information about the quality of the storage and the transport conditions during the four transfers of the samples that occurred through time. If there is no information, this should be stated. Also, the reader is referred to another paper for the sample handling; I suggest adding a more specific reference, for instance, what figure one needs to look at? Or to add that information or figure in the supplements.
There is no information about these sample transfers and storage – we will add a statement to that effect to the manuscript. What little we have been able to learn/discover is included in the Bierman et al 2024 Cryosphere paper.

Lines 186-187: please add some technical information about the energy used to scan the samples. The sample size, the geometry of the acquisition and if any filter was used. Was the reconstruction made using the NRecon software correcting for artefacts such as beam hardening?
We will add this information to the manuscript.

Line 292: perimeter is a quantity that is not robust when small grains are measured. What is the size of the grains analysed by Fiji, in number of pixels? One needs to have at least 300 pixels to have a robust measurement. See for instance, Francus and Pirard (2004). Testing for sources of errors in quantitative image analysis in P. Francus (ed.) 2004. *Image Analysis, Sediments and Paleoenvironments*. Kluwer Academic Publishers, Dordrecht, The Netherlands. The area measured in 2D slices underestimate the size of particles (same ref and many others). This should be stated somewhere in the text.
We did filter small particles when doing the image analysis. These details will be added to the manuscript.

Lines 295-298: I believe this might be problematic to compare these shape parameters taken on images with different resolutions (see again the same paper motioned above). I'm not sure what the Tukey-Kramer honestly significant difference statistical test does, and maybe it is good enough, but you should discuss a possible bias.
Thank you for this suggestion. We will consider biases in these tests used and add commentary to the manuscript.

Lines 301-313: I'm not sure to understand in what order these statistical analyses have been performed: first, first PCA on the image analysis data set on quartz grains only, and then K-means clustering on all data? Correct? Please try to clarify your text.

We will clarify this.

Lines 316-322: are these new observations, or corroborating the results of the paper cited? The wording is not clear about this.

These are new observations from the paper. We will clarify the wording on this front.

Figure 2: this figure is good-looking, and I'm sure you spent a lot of time on it. However, it needs to be improved. First, historical images are not visible in the panel a). The figure would gain in readability if the top and the bottom of the section were indicated. A scale is missing. Adding a depth scale is also needed to help read the following lines. Panel b): What is exactly the full view? Do you mean a topogram, i.e., the equivalent of a radiograph? On what is based on the colour code? And what does it mean? These are false colour, right? So what do you gain here to transform the greyscale original images to these colourful images? If you gain something, that is correct, but if not, you should consider greyscale for the CT images. Also, I think that *Climate of the Past* has a policy regarding these figures to make sure that colour-blind people can read them. Finally, what exactly is showing the *Particle view*? Have you segmented all the denser particles from the volume and made a sum of them in one direction? How have you made this segmentation? What is the smallest size of the threshold particles?

To answer this comment, we will first answer the clarifying questions included:

- What is exactly the full view? Do you mean a topogram, i.e., the equivalent of a radiograph?
    - Full view is an unaltered scan (i.e, no density filtering or lateral slicing/cropping). It goes beyond a topogram as it is a still of a reconstructed scan. We will clarify this wording
- On what is based on the colour code? And what does it mean? These are false colour, right? So what do you gain here to transform the greyscale original images to these colourful images? If you gain something, that is correct, but if not, you should consider greyscale for the CT images.
    - The color is based on density but there is no consistent density assignments to the color values, they are relative withing each core segment. We assigned the denser particles within a sample a deep red color. We think this eases interpretation as red is often attributed to higher values. In the original grey scale, the dense particles are lighter in color which could be counterintuitive to some. In addition, colorization of the grayscale density scan improves readability as the human eye can distinguish between approximately 10 millions hues compared to about 1000 gray values.

- Also, I think that *Climate of the Past* has a policy regarding these figures to make sure that colour-blind people can read them.

○ We thank the reviewer for this important insight and will provide alternative visualizations of this figure in the supporting information, including in grayscale. This is easily achievable using the imageJ "daltonize" tool which generate image adapted to protanopia, deuteranopia, tripanopia and other types of color blindness.

• Finally, what exactly is showing the *Particle view*? Have you segmented all the denser particles from the volume and made a sum of them in one direction? How have you made this segmentation? What is the smallest size of the threshold particles?

○ You are correct that "particle view" is showing dense particles at a certain threshold. This was done with the Brucker CTVox software which includes a thresholding tool. The size of particles is dependent on the resolution of the CT scanner (70 microns).

Your comments regarding readability are very useful, thank you for including them. We will consider these to improve this figure.

The labels "units 1" and "unit 2", in the 3rd column of panel a) seems to have been inverted, making the explanations below very difficult to understand.
Thank you for pointing this out. We will revise this to improve readability.

The vertical clear ice inclusions are not obvious. Are those the very narrow vertical lines, one on the left of the image, the other in the centre?
Yes those are the vertical inclusions.

Line 333: please add the name of the samples, i.e.1063-6 to 1062-1, in a similar way than you did below.
We will make this change in revision.

Line 335: what size are the clasts?
We will add this information in revision.

Line 336: sample 1062-4 seems to be a better example of these ice lenses with a braided lenticular pattern.
Yes 1062-4 does exhibit this pattern. We will consider emphasizing this sample as a good example for this pattern.

Lines 337-339: it is not clear what the reader has to look at in figure 2a that is representing ice. Where exactly are these 2 ice-rich layers?
We will add this information in revision.

Lines 340-341: it seems that the topmost sample of unit 1 is 1061-D5, right? Unless you call a "section" something that corresponds to a core tube in Bierman's paper. This is why you need to incorporate the information from the Bierman's paper about the sample names in this paper as well.

We will add this information in revision.

Line 342: I count 7 samples.

We will double check this value.

Line 343: how have you obtained this density and the % ice content? Is it with µCT? If yes, this is not trivial to obtain, and you should explain how you acquired these numbers.

This is explained in Bierman et al. which we will reference here and explain.

Lines 344-345: not all the samples display 45˚ bedding: samples 1061-D1 and 1061-D2 display horizontal contacts.

There are some artifacts from CT scanning that make some interpretations of these features difficult. We will investigate these samples to confirm the validity of these interpretations and clarify this shortcoming in the text.

Line 345: where is sample 1060-D3 in panel a)? Maybe you mean 1061-D3? If this is the case, then panel b) labelling needs to be corrected. Actually, sample 1060-D3 does not seem to exist elsewhere.

Thank you for pointing out this oversight. This will be corrected. The sample is incorrectly labeled as 1060-D3 in panel b.

Line 348: this sentence "The samples in this unit are 1060-C3, 1060-C2, and the lower portion of 1060-C1." should start the paragraph. I do not see the bedding, the grading and the cryostructures in the µCT-Scan images.

We will clarify this in the text.

Line 349: the text says here there is no bedding but the line above, there was bedding. Please review your text.

We will review this in the manuscript.

Line 351: can you better show on the picture the reticulate structure that you mention here?

We will review the samples for a better example of reticulate structures.

Line 352: bedding is visible in 1060-C1, but cryptic in 1060-C2.

Some filtering of ice/particles can reveal these features. There is a link to the entire archive that should include multiple view points of the samples for reference.

Line 353: How the ice content has been measured? (Same question as above)

This is explained in Bierman et al. which we will reference here and explain.

Lines 354-355: how can the reader know what is the a) sample and the b) sample in Figure 2? One can guess that the b) is on the right, but please add something about this in the figure.
We will review this.

Lines 355-356: Authors write "Directly below the contact bedding curves from sub-horizontal, downward to nearly 90° which continues into 1060-C", but the bedding is not visible in the picture of 1060-C2.
Good point, we will consider adding specific examples of stratigraphic features in the SI for readability.

Section 4.1: are the results presented here from the observation of the μCT scan image only?
Yes. They are supplemented by previous work in Bierman et al. 2024

Figure 3: Please change the label "14A" (standing for 14 Angstrom I suppose) from the legend into something like "clay minerals". Also, it seems that there are more amorphous minerals in samples 1060-A2 and 1059-7. Could you comment on that?
We'd rather not change the label to "clay minerals" as this would restrict interpretation. While it is true that the majority of minerals with a 14A reflection are phyllosilicates, principally in the chlorite group, other minerals can display predominant 14A d-spacing (tobermoreite or zeolites for example). We do not attribute significance to the higher level of background in the XRD of 1060-A2 and 1059-7 as we do not identify a broad "hump" at mid d-spacing value. In fact the high initial background is due to a stronger contribution of the background slide due to a smaller amount of material available for analysis in these samples.

Line 383: "selected" instead of "select", right?
We will consider this edit.

Figure 4a: what the vertical axis in the area plots means? Is it depth in the section of the sample? This is not easily readable. I would only keep the histograms. Also, in general, one plots those particle size distribution using a logarithmic scale.
We are not sure that we understand that comment, which means that our figure is indeed hard to read. We suspect that the reviewer wonders about the spread of the individual points. We will consider removing these and plotting on a logarithmic scale.

Figure 4b: this plot is not readable. I suggest making scatter plots with the size, this will be helpful to check if size influences the two other parameters.
Size is indicated by the size of each dot on the graph. We will consider this for readability.

Lines 424-425: it would be a good idea to repeat here the formula of the parameters, so the figure is self-standing.

We will add this to the figure.

Lines 434-435: I suggest adding here what group the samples belong to (for ex. A-samples,...)
We will make this change.

Line 436: "maximum mean value" is more appropriate, same for the minimum.
We will make this change.

Line 437: is roundness distribution unimodal? I really doubt it is, the spread is very wide when looking at the figure.
We will look closer into these results.

Figure 5b: is the sum of the two plots making 100%. If yes, why do you need two plots?
The sum is not 100% as it doesn't represent moderately coated grains.

Lines 448-449: see my earlier comment on potential bias of the size of the particles on the shape indices.
Thank you we will implement the discussion of bias.

Line 454: sorry, but Fig.5a does not allow to see the coatings. Fig5 is too small for that.
We will adjust the figure for readability.

Line 470: "grains within the core": which one?
We will replace these words with "grains from almost all sedimentary units" to avoid confusion.

Lines 462-471: it is a pity that there is no detailed account of all these features, to have at least a semi-quantitative view of their occurrences.
We will consider how to add this.

Section 4.3: how the analysed grains have been selected? Random selection is quite important to avoid biases, or maybe all visible grains were selected.
Random selection was used, we will add this to the method description.

Figure 6: the figure is very nice, but the element code in panel g is not clear and the scale shows the ∞ character, I suppose instead of the µ one.
We will correct this.

Figure 7c: the variable labels are too small.
We will correct this.

Line 495: why have you included depth as a variable? I suppose it is the depth in the core. If it is, then I think this is biasing your statistical analysis, forcing the samples with the same depth to be similar (spatial autocorrelation). I think you should remove this variable, and redo your statistical analysis without that variable. This brings the question how the variables included in the analysis have been selected? Can you expand on that?

We used all the data we could for this analysis as it is exploratory. As depth is relevant in stratigraphy and interpretations of stratigraphic deposits, we elected to retain depth as a variable so as to include as much information as possible into the algorithm.

Line 506: in the introduction, you write that the sediment core is made of several units, but here you assess this unit assignment. From this, the reader understands that the units have been previously defined, and I suppose this was made in previous papers. The authors should clearly distinguish the interpretation derived from the dataset presented here from the other proxy not presented here. Also, could you suggest a change in the unit assignments using your dataset only?

What is new here and different from Christ et al. (2021) and Bierman et al. (2024) is the thorough and in depth stratigraphic characterization based not only on visual observation but on numerous parameters measured at a variety of scales. While previous work identified units macroscopically, here we assess the microscopic, sedimentologic, and mineralogical nature of these units and propose a scenario for the past evolution of the ground surface environment beneath Camp Century. We will as the reviewer suggests point out this fundamental difference in the revised text more clearly and provide specific details about how what we have done in this study differs from studies that came before.

Line 530: I suggest adding "subsequent" here : (…) before subsequent cooling (…)

We will correct this sentence.

Line 535: which cryostructure are you talking about?

Will add specifics.

Line 536: this is the first time you mention that slumping is occurring. You should first demonstrate that the sediment is indeed a slump.

This came up in the other review and we will address by providing additional data consistent with slumping.

Lines 538-539: slumps can also produce debris flows, which are sediment facies that are not well sorted. If there is a normal grading, it is more likely that a turbiditic current occurred, implying that the environment is not compatible with sediments flowing downslope. Many questions come to mind here: for instance, was the site in an aerial or aqueous environment?

Similar questions were raised by the other reviewer, using all data available to us, we will reconsider our interpretation and terminology and provide a more rigorous analysis of this unit.

Lines 535-544: this interpretation for unit 3 is very hypothetical. I don't think that these inferences can be made out of a single core. One needs to have observations about what is happening laterally.

There are no lateral observations as this is a drill core but we do have other data indicating very strongly that unit 3 was derived at least in part from unit 1 and that it is a poorly sorted diamict. During revision we will make a more reasoned argument with supporting data.

Line 556: sediment content: what do you refer to here? Grain size? The horizontal alignment of grains is present only in a few samples.

We are referring to the suspended grains in the ice matrix in Unit 2. The wording will be revised for clarity.

Line 562: fluvial sediments are usually better sorted that these.

As the degree of sorting in fluvial systems is dependent (among other parameters) on the energy of the system and size of the stream we think that this term is applicable to a small periglacial stream.

Line 565: "Multiple lines of evidence": please specify which ones. The ones from this dataset or from other papers? Your dataset is not very convincing that this unit has been created by a river.

We will be more specific in citing data sets supporting our rationale.

Line 569: is there information about the topography under the ice cap?

Yes. There is a large radar dataset and we will report on it but it is at a scale much too coarse to evaluate site specific topography.

Line 591: this is counter-intuitive: glacial tills are usually coarser than fluvial sediments.

We will consider this and add citation to back our findings.

Figure 8: the font size in the boxes are too small.

We will correct this.

---

## Author Response (AR1)

**Review 1**

The paper by Catherine Collins and co-authors presents a new rich dataset from the precious sediment samples collected at the base of the Camp Century ice core. The analyses include microCT, XRD, SEM-EDS and SEM imaging. The authors looked at the sedimentary facies in order to decipher the processes responsible for the deposition and preservation of sediments and infer some paleoenvironmental interpretation. The paper is well written and the figures have been made with great care, although several improvements can be done (see detailed comments below).

We appreciate reviewer's enthusiasm for our work.

This paper refers a lot to another one submitted to the journal *The Cryosphere* by Bierman *et al.* (still a preprint at the time this review is written).

This paper is now published and we have updated references.

Unfortunately, the reader needs to read the Bierman *et al.* paper to understand several things, such as the sample numbering and the stratigraphy. I think this essential information needs to be incorporated within this paper. For instance, the uppermost core section is the one on the right in the figure, which is not what one usually when presenting sediment cores. Incorporating panel B of figure 2 of the Bierman's paper would be very helpful.

We have completely reworked this figure which now includes the 3D photograms from Bierman et al. 2024. If the editor insists, we can replicate the figure but that seems un-necessary.

A disturbing feeling while reading the discussion section of this paper is that many arguments for the interpretations of the sediment facies are coming from previous papers. A clear distinction between the interpretations based on this new dataset, and the data presented elsewhere is needed.

We have revised discussion extensively to focus on the new data so that interpretations are clearly distinct from those of prior paper.

My main concern is about the interpretation of the sedimentary facies. I'm not a specialist of glacial/proglacial/subglacial microfacies, but in such settings, it is critical to have an idea of position of the ice, the direction of the flows, the topography, the slopes, etc., to make a solid interpretation, which is impossible to obtain from a single sediment core.

We agree that our ability to make interpretations is limited because of the size of the core and our inability to place what we see in a broader landscape context. In particular, the core was never oriented and the sub-core stored without relative orientation. We added this information in the methods L186. Neverthe less, there are robust interpretations we can and have made from the now large amounts of data available for this core.

Moreover, it is very likely that the sediments recovered at the bottom of 1350 m of ice experienced some glaciotectonism. This is not discussed in the manuscript, and it should, because it is not sure that the ice cap was cold based all the time, especially if one of the interpretations suggested by the authors is that a river flowed at this site. In short, I think that the observations remain interesting but that the facies interpretation is pushed too far. It would be

useful to have the input of a specialist of microsedimentological analyses of glaciogenic sediments.

We appreciate the comment and clarified the link between our observation and interpretation in the text L600. We also cite Larson et al 2016 and Menzies and Reitner 2016 to support our interpretation. Others are conducting more detailed analyses of the CT scans which may provide more specific data based on clast orientation but that work is in progress and outside the scope of this particular paper. We believe the thermal state of the ice sheet and the presence of moving water when the area was deglaciated are two independent observations.

For the statistical interpretation, the choice of the variables considered is not well justified; for instance, why including the depth of the sample as a variable (it could be an explanatory data).

We may not fully understand the second part of the reviewer's question - "why it couldn't have been an explanatory variable?" We did use depth as one of our predictors (explanatory) variables. We have expanded our justification for variable choices.

Finally, there are many small mistakes and inaccuracies in the μCT dataset description making the paper arduous to read. There are details about the μCT data acquisition in the online public repository (congratulations for that effort), but it would be nice adding some information in the paper and/or in the repository in order to be able to reproduce the measurements (see my detailed comments below).

We have tried to remove mistakes, expand and clarify the CT description.

Unfortunately, I was unable to open all the movies made on processed images.

We will investigate with the repository why this might be the case. They work for us.

While the movies are a very nice addition, it is quite impossible to analyse these images. The addition of images that can be handled by an image analysis software would have been better in my point of view, but this is already very nice.

The raw SEM and CT scan data is currently in review by the artic data center and should be accessible soon.

**Detailed comments:**
Line 65 : « sequence stratigraphy » has a specific meaning in sedimentology that is not related to the reality you want to describe. I suggest changing this by "the lithological succession".

Agree.

Line 166: "shrinking as sediment freezes". I'm not a specialist of these structures, but it does not seem logical to have shrunk when ice has a 10% higher volume. Can you expand on that?

Thank you for pointing this out. As written this sentence excludes part of what is included in French & Shur, 2010 which is explained as follows: "Reticulate cryostuctures are thought to reflect desiccation and shrinkage as sediment progressively freezes and moisture migrates to the advancing freezing front (e.g., Mackay, 1974)." We added this citation and clarified the sentence to include this nuance.

Lines 172-173: this statement is strong and should be supported by evidence or a reference to other studies.
This has been edited.

Lines 178-182: these sentences belong to the introduction, not the method section.
We will move these sentences.

Method section: there is little information about the quality of the storage and the transport conditions during the four transfers of the samples that occurred through time. If there is no information, this should be stated. Also, the reader is referred to another paper for the sample handling; I suggest adding a more specific reference, for instance, what figure one needs to look at? Or to add that information or figure in the supplements.
There is no information about these sample transfers and storage – we will add a statement to that effect to the manuscript. What little we have been able to learn/discover is included in the Bierman et al 2024 Cryosphere paper.

Lines 186-187: please add some technical information about the energy used to scan the samples. The sample size, the geometry of the acquisition and if any filter was used. Was the reconstruction made using the NRecon software correcting for artefacts such as beam hardening?
We have added this information to the manuscript.

Line 292: perimeter is a quantity that is not robust when small grains are measured. What is the size of the grains analysed by Fiji, in number of pixels? One needs to have at least 300 pixels to have a robust measurement. See for instance, Francus and Pirard (2004). Testing for sources of errors in quantitative image analysis in P. Francus (ed.) 2004. *Image Analysis, Sediments and Paleoenvironments*. Kluwer Academic Publishers, Dordrecht, The Netherlands. The area measured in 2D slices underestimate the size of particles (same ref and many others). This should be stated somewhere in the text.
We did filter small particles when doing the image analysis. These details are added to the manuscript.

Lines 295-298: I believe this might be problematic to compare these shape parameters taken on images with different resolutions (see again the same paper motioned above). I'm not sure what the Tukey-Kramer honestly significant difference statistical test does, and maybe it is good enough, but you should discuss a possible bias.
Thank you for this suggestion. We considered biases in these tests used and add commentary to the manuscript.
Lines 301-313: I'm not sure to understand in what order these statistical analyses have been performed: first, first PCA on the image analysis data set on quartz grains only, and then K-means clustering on all data? Correct? Please try to clarify your text.
The PCA is an integral part of the K-means clustering. By first reducing the dimensionality of the data via PCA, applying the K-means clustering reveals group of similar behaviors (see Ding and He, 2004 – "K-means Clustering via Principal Component Analysis"). So the PCA was performed on everything. We clarified that adding "We performed a PCA to create a new set of uncorrelated variables to help reduce the noise in our data, and then performed a K-means

clustering" which should also help clarify the order of actions.

Lines 316-322: are these new observations, or corroborating the results of the paper cited? The wording is not clear about this.
These are new observations from the paper.

Figure 2: this figure is good-looking, and I'm sure you spent a lot of time on it. However, it needs to be improved. First, historical images are not visible in the panel a). The figure would gain in readability if the top and the bottom of the section were indicated. A scale is missing. Adding a depth scale is also needed to help read the following lines. Panel b): What is exactly the full view? Do you mean a topogram, i.e., the equivalent of a radiograph? On what is based on the colour code? And what does it mean? These are false colour, right? So what do you gain here to transform the greyscale original images to these colourful images? If you gain something, that is correct, but if not, you should consider greyscale for the CT images. Also, I think that *Climate of the Past* has a policy regarding these figures to make sure that colour-blind people can read them. Finally, what exactly is showing the *Particle view*? Have you segmented all the denser particles from the volume and made a sum of them in one direction? How have you made this segmentation? What is the smallest size of the threshold particles?
We reworked figure 2 to include the reviewer's comment: this way we address the issue of color coding and describe the various "views" better in the methods and caption.

- What is exactly the full view? Do you mean a topogram, i.e., the equivalent of a radiograph?
    - Full view is an unaltered scan (i.e, no density filtering or lateral slicing/cropping). It goes beyond a topogram as it is a still of a reconstructed scan.
- On what is based on the colour code? And what does it mean? These are false colour, right? So what do you gain here to transform the greyscale original images to these colourful images? If you gain something, that is correct, but if not, you should consider greyscale for the CT images.
    - The color is based on density but there is no consistent density assignments to the color values, they are relative withing each core segment. We assigned the denser particles within a sample a deep red color. We think this eases interpretation as red is often attributed to higher values. In the original grey scale, the dense particles are lighter in color which could be counterintuitive to some. In addition, colorization of the grayscale density scan improves readability as the human eye can distinguish between approximately 10 millions hues compared to about 1000 gray values.

- Also, I think that *Climate of the Past* has a policy regarding these figures to make sure that colour-blind people can read them.
    - We thank the reviewer for this important insight and will provide alternative visualizations of this figure in the supporting information, including in grayscale. This is easily achievable using the imageJ "daltonize" tool which generate image adapted to protanopia, deuteranopia, tripanopia and other types of color blindness.

- Finally, what exactly is showing the *Particle view*? Have you segmented all the denser particles from the volume and made a sum of them in one direction? How have you made this segmentation? What is the smallest size of the threshold particles?
  - You are correct that "particle view" is showing dense particles at a certain threshold. This was done with the Brucker CTVox software which includes a thresholding tool. The size of particles is dependent on the resolution of the CT scanner (70 microns).

The labels "units 1" and "unit 2", in the 3rd column of panel a) seems to have been inverted, making the explanations below very difficult to understand.
Thank you for pointing this out. We have revised to improve readability.

The vertical clear ice inclusions are not obvious. Are those the very narrow vertical lines, one on the left of the image, the other in the centre?
Yes those are the vertical inclusions.

Line 333: please add the name of the samples, i.e.1063-6 to 1062-1, in a similar way than you did below.
We will made this change

Line 335: what size are the clasts?
We added "from sand to cobble size" to the text

Line 336: sample 1062-4 seems to be a better example of these ice lenses with a braided lenticular pattern.
Yes 1062-4 does exhibit this pattern. As now shown on figure 2, all samples from Unit 1 show such patterns

Lines 337-339: it is not clear what the reader has to look at in figure 2a that is representing ice. Where exactly are these 2 ice-rich layers?
We reworked this sentence to read: The transition is not discrete as the volume of ice increases upward within the fine-grained matrix material of Unit 1.

Lines 340-341: it seems that the topmost sample of unit 1 is 1061-D5, right? Unless you call a "section" something that corresponds to a core tube in Bierman's paper. This is why you need to incorporate the information from the Bierman's paper about the sample names in this paper as well.
Both 1062-1 and 1061-D5 are constitutive of the upper part of Unit 1. In fact we consider that the transition occurs within 1061-D5. We now clarified in the text.

Line 342: I count 7 samples.
Correct

Line 343: how have you obtained this density and the % ice content? Is it with $\mu$CT? If yes, this is not trivial to obtain, and you should explain how you acquired these numbers.
We reference to Bierman et al 2024 here

Lines 344-345: not all the samples display 45˚ bedding: samples 1061-D1 and 1061-D2 display horizontal contacts.

We confirm the validity of this statement which can be verified using the raw data and unfortunately is hard to show in paper form.

Line 345: where is sample 1060-D3 in panel a)? Maybe you mean 1061-D3? If this is the case, then panel b) labelling needs to be corrected. Actually, sample 1060-D3 does not seem to exist elsewhere.

This has been corrected

Line 348: this sentence "The samples in this unit are 1060-C3, 1060-C2, and the lower portion of 1060-C1." should start the paragraph. I do not see the bedding, the grading and the cryostructures in the µCT-Scan images.

We reworded as suggested and clarified the text. The new Figure 2 should also help visualize the samples .

Line 349: the text says here there is no bedding but the line above, there was bedding. Please review your text.

We modified the text (this comment and the next 4 comments)

Line 351: can you better show on the picture the reticulate structure that you mention here?
We will review the samples for a better example of reticulate structures.
I also am not sure what that is. Cat an idea? Or we remove.

Line 352: bedding is visible in 1060-C1, but cryptic in 1060-C2.
Some filtering of ice/particles reveals these features. There is a link to the entire archive that includes multiple view points of the samples for reference.

Line 353: How the ice content has been measured? (Same question as above)
This is explained in Bierman et al. 2024

Lines 354-355: how can the reader know what is the a) sample and the b) sample in Figure 2? One can guess that the b) is on the right, but please add something about this in the figure.
We will review this.

Lines 355-356: Authors write "Directly below the contact bedding curves from sub-horizontal, downward to nearly 90° which continues into 1060-C", but the bedding is not visible in the picture of 1060-C2.
We clarified this in the text. The new Figure 2 shows this well.

Section 4.1: are the results presented here from the observation of the µCT scan image only?
Yes. They supplement previous work in Bierman et al. 2024

Figure 3: Please change the label "14A" (standing for 14 Angstrom I suppose) from the legend into something like "clay minerals". Also, it seems that there are more amorphous minerals in samples 1060-A2 and 1059-7. Could you comment on that?

We'd rather not change the label to "clay minerals" as this would restrict interpretation. While it is true that the majority of minerals with a 14A reflection are phyllosilicates, principally in the chlorite group, other minerals can display predominant 14A d-spacing (tobermoreite or zeolites for example). We do not attribute significance to the higher level of background in the XRD of 1060-A2 and 1059-7 as we do not identify a broad "hump" at mid d-spacing value. In fact, the high initial background is due to a stronger contribution of the background slide due to a smaller amount of material available for analysis in these samples. We added a sentence in the method to that effect.

Line 383: "selected" instead of "select", right?

Now deleted

Figure 4: We reworked figure 4 to display the particle size distribution in log scale and the data spread as violin. We kept figure 4b as is as we think that the figure is bringing more information this way than simple scatterplots.

Figure 4a: what the vertical axis in the area plots means? Is it depth in the section of the sample? This is not easily readable. I would only keep the histograms. Also, in general, one plots those particle size distribution using a logarithmic scale.

We are not sure that we understand that comment, which means that our figure is indeed hard to read. We suspect that the reviewer wonders about the spread of the individual points. We will consider removing these and plotting on a logarithmic scale.

Figure 4b: this plot is not readable. I suggest making scatter plots with the size, this will be helpful to check if size influences the two other parameters.

We have tried to increase readability by redrafting figure.

Lines 424-425: it would be a good idea to repeat here the formula of the parameters, so the figure is self-standing.

We are not sure what formula that is.

Lines 434-435: I suggest adding here what group the samples belong to (for ex. A-samples,…)

Letters added

Line 436: "maximum mean value" is more appropriate, same for the minimum.

Change made.

Line 437: is roundness distribution unimodal? I really doubt it is, the spread is very wide when looking at the figure.

Figure 4a clearly indicates unimodality in distribution for roundness.

Figure 5b: is the sum of the two plots making 100%. If yes, why do you need two plots?

The sum is not 100% as it doesn't represent moderately coated grains.

Lines 448-449: see my earlier comment on potential bias of the size of the particles on the shape indices.
We added this aspect to the methods

Line 454: sorry, but Fig.5a does not allow to see the coatings. Fig5 is too small for that.
We disagree.

Line 470: "grains within the core": which one?

Lines 462-471: it is a pity that there is no detailed account of all these features, to have at least a semi-quantitative view of their occurrences.
This will come in a later paper.

Section 4.3: how the analysed grains have been selected? Random selection is quite important to avoid biases, or maybe all visible grains were selected.
Random selection was used, added this to the method description.

Figure 6: the figure is very nice, but the element code in panel g is not clear and the scale shows the ∞ character, I suppose instead of the μ one.
corrected.

Figure 7c: the variable labels are too small.
corrected

Line 495: why have you included depth as a variable? I suppose it is the depth in the core. If it is, then I think this is biasing your statistical analysis, forcing the samples with the same depth to be similar (spatial autocorrelation). I think you should remove this variable, and redo your statistical analysis without that variable. This brings the question how the variables included in the analysis have been selected? Can you expand on that?
We used all the data we could for this analysis as it is exploratory. As depth is relevant in stratigraphy and interpretations of stratigraphic deposits, we elected to retain depth as a variable so as to include as much information as possible into the algorithm.

Line 506: in the introduction, you write that the sediment core is made of several units, but here you assess this unit assignment. From this, the reader understands that the units have been previously defined, and I suppose this was made in previous papers. The authors should clearly distinguish the interpretation derived from the dataset presented here from the other proxy not presented here. Also, could you suggest a change in the unit assignments using your dataset only?
What is new here and different from Christ et al. (2021) and Bierman et al. (2024) is the thorough and in depth stratigraphic characterization based not only on visual observation but on numerous parameters measured at a variety of scales. While previous work identified units macroscopically, here we assess the microscopic, sedimentologic, and mineralogical nature of

these units and propose a scenario for the past evolution of the ground surface environment beneath Camp Century. We have as the reviewer suggested pointed out this fundamental difference in the revised text more clearly and provide specific details about how what we have done in this study differs from studies that came before.

Line 530: I suggest adding "subsequent" here : (…) before subsequent cooling (…)
corrected

Line 535: which cryostructure are you talking about?
Added specifics.

Line 536: this is the first time you mention that slumping is occurring. You should first demonstrate that the sediment is indeed a slump.
We provide additional data showing that unit 3 is consistent with slumping.

Lines 538-539: slumps can also produce debris flows, which are sediment facies that are not well sorted. If there is a normal grading, it is more likely that a turbiditic current occurred, implying that the environment is not compatible with sediments flowing downslope. Many questions come to mind here: for instance, was the site in an aerial or aqueous environment?
Similar questions were raised by the other reviewer, using all data available to us, we have tried to more clearly explain the origin of unit 3.

Lines 535-544: this interpretation for unit 3 is very hypothetical. I don't think that these inferences can be made out of a single core. One needs to have observations about what is happening laterally.
There are no lateral observations as this is a drill core but we do have other data indicating very strongly that unit 3 was derived at least in part from unit 1 and that it is a poorly sorted diamict. During revision we made a more reasoned argument with supporting data.

Line 556: sediment content: what do you refer to here? Grain size? The horizontal alignment of grains is present only in a few samples.
We are referring to the suspended grains in the ice matrix in Unit 2. Wording revised for clarity.

Line 562: fluvial sediments are usually better sorted that these.
As the degree of sorting in fluvial systems is dependent (among other parameters) on the energy of the system and size of the stream we think that this term is applicable to a small periglacial stream.

Line 565: "Multiple lines of evidence": please specify which ones. The ones from this dataset or from other papers? Your dataset is not very convincing that this unit has been created by a river.
We are no more specific in citing data sets supporting our rationale.

Line 569: is there information about the topography under the ice cap?
Yes. There is a large radar dataset but it is at a scale much too coarse to evaluate site-specific topography.

Line 591: this is counter-intuitive: glacial tills are usually coarser than fluvial sediments.
This is not always the case, depends completely on source materials and fluvial sorting.

Figure 8: the font size in the boxes are too small.
Corrected

**Review 2**

Collins et al. (2024) present a comprehensive multiscale analysis of the sub-glacial core collected from Camp Century, Greenland, analyzing physical, geochemical, and cryostratigraphic characteristics. The authors use X-ray diffraction (XRD), micro-computed tomography (μCT), and scanning electron microscopy (SEM), to identify and characterize five distinct sedimentary units, each with characteristic structures, mineralogical compositions, and cryostructures, and then use these data to infer a sequence of depositional environments, spanning glacial to interglacial to glacial. Their findings add to a growing body of work on this subglacial material and provide further context into Greenland's paleoclimate history and ice sheet dynamics, though some of their interpretations remain speculative without further supporting evidence.

The reviewer accurately captured the nature and aims of our work. We agree that our interpretations are limited to the evidence we have.  Nevertheless, as described in the points below, we will carefully assess the level of speculation within our interpretation.

Overall, this paper is well-written and polished, and the data and generally analyses are robust. However, my primary critique of this work is that the authors' interpretations occasionally overreach given the data presented and some aspects of the presentation lack sufficient detail to clearly follow their logic. While the authors' interpretations are plausible, it appears they could represent one possible scenario among several.

This is a reasonable critique and we  worked  in revision to be more explicit about what data we have and what suggestions are more certain and which are more speculative.
The reviewer will find detail of this below.

This is particularly uncertain when inferring the timing of cryostructure formation or depositional environments from qualitative observations. I believe the paper provides a solid empirical foundation and presents a valuable dataset but lacks specific evidence for some interglacial interpretations.

We have significant additional data to support interglacial age assignments (see below) but no firm data on the age of cryostructures and now state that clearly.

I describe my main concerns with the interpretation of the sediment facies below and then follow up with minor concerns.

"The higher ice content and the difference in cryostructures between Units1 and 3 supports the hypothesis that Unit 3 was deposited in or by liquid water."
The specific differences in cryostructures are not clearly presented in this manuscript. The paper lacks plots or tables that detail ice content by depth, as well as quantitative descriptions of ice lens characteristics such as shape, spacing, orientation, and thickness.

Agree, we will did our best to provide more of these type of data in revision and reduce our emphasis on these structures.

These details could provide valuable clues to the origin of the ice lenses and would allow for comparisons with theoretical predictions. Including a figure (in the supplemental material if

necessary) that details and compares the various cryostructures within each unit in high resolution would greatly enhance clarity, as these structures are frequently referenced as evidence for interpreting the origins of each unit. Currently, it is challenging to contextualize the manuscript's claims alongside the physical evidence. The supplemental movies provide a helpful visual, but lack sufficient detail to quickly orient the reader.

This is an excellent suggestion we did our best to incorporate specific details of cryostructures from the CT scans. We have now revised the description of the CT scans, modified figure 2 for ease of reading and adding a new figure to illustrate specific structures of the core.

(As a note, I was unable to view the CT scan videos using an M1 Mac on either Safari or Firefox and ultimately had to download Chrome, which played them successfully.)

We deposited the scans in an NSF-supported repository (under review) and I have downloaded them in Safari on Intel, Firefox on PC and M2 Mac – not sure what the difficulty might be but not in our control.

**Regarding the Interpretation for Unit 2**
The authors suggest that Unit 2 is a ~1-m-thick remnant of basal ice (or even firn) preserved from a previous glaciation that survived interglacial conditions near the surface (e.g., Figure 8). This seems challenging to support without corroborating evidence, such as ice crystallography or age constraints.

We have neither ice crystallography nor any robust age constraints on the unit 2 but, initial luminesence data, neither complete or public, suggests it is similar in age to the material above it (400-450 ky). The notion of firn/basal ice is supported by unpublished stable water isotope data that are part of MS thesis. We have revised integrate possible alternative as these other data re too broad to be integrated within this work and currently the subject of articles in preparation

The resemblance to "basal ice" is unsurprising, given that this layer is located in subglacial frozen sediments near the ice-bed interface, akin to a frozen fringe. Theory predicts that a frozen fringe could grow meters thick and form ice lenses in basal till under freezing conditions (e.g., Meyer et al., 2023), following similar physics as frost heave in permafrost—an occurrence well-documented in the field (e.g., Christofferson and Tulacyzk, 2003; Fitzsimmons et al., 2024). A recent paper posits that clasts can migrate into ice through thermal regelation (Pierce et al., 2024), which could provide a mechanism to progressively incorporate till from the underlying layer. Ice crystal structure analysis could clarify whether this is highly sheared basal ice left over from a previous glaciation or ice grown through cryosuction, which would exhibit distinctive characteristics. Without such data, the timing of its emplacement remains speculative. One might imagine Unit 3 being deposited over Unit 1, with later ice lens growth. Based on the information provided, I don't see how it could be said with certainty.

The reviewer is correct – without direct geochronology we cannot for certain pinpoint the age of unit 2. Pore ice pH and conductivity of the core (Bierman et al 2024) show that they are dramatically different geochemically.

Additionally, while the presence of vertical cracks is intriguing and could potentially represent freeze-thaw cycles, I do not believe they are a smoking gun. This layer could have experienced any number of glaciotectonic events that induced brittle strain, and the history is difficult to deduce from a single core, which is essentially a point source.

We agree that freeze-thaw is not the only possible explanation, and have added potential for glacial induced stress and cracking in the discussion.

**Regarding the Interpretation for Unit 3**
The authors describe Unit 3 as a slump deposit, partly based on the presence of normal grading. This interpretation is not readily apparent to me from the evidence provided in Figure 2. The core sections labeled as Unit 3 are divided between 1060-C3 and the lower portion of 1060-C1, with a substantial gap between them, which complicates a continuous interpretation. In the lower section (1060-C3), the sample is clast-rich, while the upper portion (1060-C1) contains mostly fines. This contrast presumably forms the basis for describing Unit 3 as normally graded. However, 1060-C3 does not visually appear graded; it lacks the clear stratification expected in a slump deposit and could simply represent basal till, with clasts suspended in a finer matrix. We confirmed the evidence for normal grading and added to the revised ms additional data from other sources that also suggest this is a slump deposit. This portion of the core in in fact very complete and continuous as shown on figure 2 with the ct-scans of subcores 1060-C3, 1060-C2 and 1060-C1 in direct connection. Changes made to figure 2 and the added SI figure should resolve the reviewer's suggestion. In particular, the addition of 1060-C4 to the figure.

The missing sediment between 1060-C3 and 1060-C1 leaves some ambiguity in interpreting the contact between these layers. This discontinuity between a poorly sorted till and fines could indicate separate depositional events/mechanisms, weakening the argument for a continuous slump feature. Additional data—such as grain size distributions with depth or clast fabric/orientation relative to underlying till sections—would lend stronger support to the slump hypothesis (though it would still be difficult to ascertain from a single core without lateral context). Clast orientation specifically could help distinguish a traction till, characterized by horizontal shear, from a flow till, deposited downslope. In lieu of these constraints, alternative interpretations are plausible.
All of these ideas are wonderful but with a several inch wide core, not possible to test. We worked to provide viable alternative explanations while continuing to focus on the interpretations we consider best supported by our observations and data. We also point out that 1060-C2 is not missing and that the sample was analyzed by XRD, SEM and ctscan.
We suspect that the reviewer wished to point to the missing sample between 1060-C1 and 1060-A2 rather than 1060-C3 and C1. 1060-C2 is presented and shows remarkable continuity with its lower and upper neighbors (both the angling of the slump and its composition). We however agree that there is a level of speculation to our interpretation and have reworded our interpretation to reflect this, now providing alternative explanations inspired by the reviewer. "While we posit that the origin of this unit is likely a flow slump, we also posit that in the absence of orientation of grains, limited lateral context and size of the cores, alternative genetic origins for unit 3 cannot be discarded, such as partial melt and current reworking of basal till. And "The paucity of grain coatings in Unit 3 compared to Unit 1 (Fig. 5) could indicate that the slumping process disrupted grain coatings or mixed sediment from Unit 1 with sediment from the upper units, 4 and 5."

If Unit 3 was deposited as a basal till and subsequently covered by fluvial deposits, then the lack of grain coatings might be expected. My impression is that the main evidence for interpreting Unit 3's deposition as subaerial or subaqueous, rather than glacial, is the presence of weathered

coatings in Unit 1 below, which suggests significant exposure at Earth's surface. This would imply that ice retreated, exposing Unit 1, before Unit 3 was subsequently deposited. An alternative hypothesis could be an ice readvance that deposited Unit 3. While the paleo-climatic sequence for this specific region isn't my specialty, are there regional constraints that would support this specific locale being ice-free for the full duration of the interval given by the age constraints of Units 1 and 5, or is this largely unknown?

We clarified our description. We don't consider a re-advance as a likely source of a very thin (dm) layer of diamicton. Until our work, there was no understanding of the paleoclimatic sequence for Camp Century area because it's been buried by ice since 400 ky. Overall, we will better state and review what chronologic constraints exist for the sub-ice material. We added "The sub-horizontal, braided, lenticular cryostructures are consistent with syngenetic permafrost formation, suggesting based on cryostratigraphy, little influence of liquid water in Unit 1 after permafrost formation (French & Shur, 2010)."

Based on the evidence provided in this manuscript, the timing of these lenticular cryostructures remains ambiguous. While the authors suggest they formed as syngenetic permafrost in a subaerial environment, it is equally plausible that they developed subglacially over a range of timescales. Without additional dating or contextual evidence, attributing them definitively to subaerial permafrost formation is speculative.

Agree, we will provide this alternative and made it clear there are other possible processes that could have formed the structures. We now have added this sentence: "While we posit that the origin of this unit is likely a subaerial flow slump, we also posit that in the absence of orientation of grains, limited lateral context and size of the cores, alternative genetic origins for unit 3 cannot be discarded, such as partial melt and reworking of nearby till by currents or subglacial stream collapse. The presence of vegetal remains in all samples (Christ et al. 2023) however suggest a subaerial context."

"Overall, the similar sedimentary structures and mineralogy suggest that Unit 3 was originally a part of the subglacial till formed below the ice (Unit 1) that was subject to slumping due to saturation of liquid water during interglacial conditions."

While I concur that this likely originated as subglacial till, the assertion that it is a slump deposit in an interglacial seems rather uncertain. As mentioned above, the timing here could have other possibilities. Even as a slump, perhaps it could be subglacial as well, perhaps representing the collapse of a canal's sidewalls.

We are now  more specific in the logic we use to suggest the origin of unit 3 and cite other data. For example, a yet to be published luminescence ages on unit 3 show that it too was exposed at MIS 11 making a subglacial origin unlikely.  We do not see a slump as fundamentally different from a channel collapse. We interpret this comment as the reviewer suggesting that the formation of unit 3 may be subglacial rather than subaerial but the luminescence age argues against that interpretation.

"Water and vegetation: During interglacial conditions, the permafrost landscape was subject to freeze-thaw cycles as shown by the vertical lenses of clear ice in Unit 2. Till, saturated by water, flowed downslope and buried the ice forming Unit 3. Interglacial conditions supported plant growth and the development of a headwaters fluvial system that eroded the upper portion of the

flow-till deposit, stripping grain coatings but not changing grain shape. The fluvial system then deposited bedded sand, initially fine-grained and then coarser-grained material."

The evidence presented here doesn't necessarily indicate a unique interglacial, subaerial depositional environment:

1. Ice lenses could have formed in these layers subglacially over an ambiguous range of timescales.
2. Unit 2 may have grown after the deposition of Unit 3. If Unit 3 represents a slump (though this is not convincingly demonstrated), it's plausible that it collapsed as a subglacial feature, potentially forming canal walls.
3. The introduction (Lines 60-62) states the presence of plant and invertebrate fossils in the core necessitates the site was ice-free during MIS11, but the authors do not explicitly state what units described in this paper contained signs of biological life. Fossils in Units 2-5 would significantly bolster the interglacial deposition argument, yet this isn't explicitly discussed in this manuscript. In particular, are there biological remains in Unit 3?

There is a lot to unpack here – some of which we can address with certainty, some we have other data that helps constrain the stratigraphy, and the rest of which will, due to lack of data, remain uncertain. For example, we have found biological remains in every sample from the core – this is the focus of an upcoming paper. There are number of geochemical and isotopic measures showing the close similarity of units 1 and 3. But, unit 2 remains poorly understood with some suggestions from stable water isotopes that it could be firn deposited on the underlying unit 1 and then buried by unit 3. In revision, we added information about organic material requested by the reviewers and where appropriate discussed alternative hypotheses and where possible, presented other data in support of our hypotheses.

**Line 172**: What evidence is there that this has undergone minimal deformation? Line 348 mentions deformed bedding, for instance, with no further explanation. Has previous work assessed the microstructure in detail? A citation would be warranted here.

This work is the first attempt at a partial microstructural analysis of the core using ct-scan. The evidence is visual. We now state this explicitly and add a reference to previous work when referring to the macrostructural analysis in this method section. We believe that the new figures now address this comment.

**Line 316**: "Our multiscale analysis supports, refines, updates, and provides more justification and detail regarding unit delineations proposed earlier (Bierman et al., 2024; Christ et al., 2021, 2023; Fountain et al., 1981)." It would be helpful to identify in the discussion how observations made in this paper agree or disagree with past interpretations of the units.

Done.

**Line 322**: Technically, it is all subglacial material. Does this refer to Units 1 and 2?
We have clarified wording

**Minor punctuation errors**

- **Line 47**: "1969)), but the sub-glacial material" → missing comma.

- **Line 172**: "minimally, if at all, deformed."

Corrected.

Works cited

Meyer, C.R., Schoof, C. and Rempel, A.W., 2023. A thermomechanical model for frost heave and subglacial frozen fringe. *Journal of Fluid Mechanics*, *964*, p.A42.

Pierce, E., Overeem, I. and Jouvet, G., 2024. Modeling sediment fluxes from debris-rich basal ice layers. *Journal of Geophysical Research: Earth Surface*, *129*(10), p.e2024JF007665.

Christoffersen, P. and Tulaczyk, S., 2003. Response of subglacial sediments to basal freeze-on 1. Theory and comparison to observations from beneath the West Antarctic Ice Sheet. *Journal of Geophysical Research: Solid Earth*, *108*(B4).

Fitzsimons, S., Samyn, D. and Lorrain, R., 2024. Deformation, strength and tectonic evolution of basal ice in Taylor Glacier, Antarctica. *Journal of Geophysical Research: Earth Surface*, *129*(4), p.e2023JF007456.

**Review 3**

Responses are in red

This is a solid paper from Paul Bierman's group on the surprising discovery of the sub-ice sediment from the Camp Century ice core recovered several decades ago and presumed lost. This follows earlier papers by Christ et al (2023) and Bierman et al (2023, 2024). Below are some specific and general comments on the ms.
Thank you Giff. The papers are Christ et al (2021, 2023) and Bierman et al (2024).

Abstract: Because more people read abstract than ms, put into the Abstract your key findings: in situ MIS 12 till overlain by MIS 12 basal debris, then a sequence of MIS 11 fluvial sediment. You want to be sure that the casual abstract reader has caught your main conclusions, which may lead them to actually read the paper.
We agree with placing key findings in the abstract and have reviewed what we have written to do that. We do not however have any chronologic control on the till or the ice layer except that cosmogenic 26Al/10Be ratios indicate that it must have been near the surface within the last 3 My. IRSL data suggest that the till has not been exposed to sunlight within the last 1.7 My (Christ et al. 2021). As we have therefore no evidence that the till is MIS 12, we can only refer to this material as "older".

Introduction: Line 1 of Intro is misleading "Understanding past ice-free times allows us to predict the response of the Greenland Ice Sheet (GrIS) to current and future climate warming (Gemery & López-Quirós, 2024)" This is not a claim in the cited article, and certainly not true as stated. "current and future climate warming" has no analog in the past in terms of rates of GHG change. This complaint in no way is meant to diminish the importance of paleo work, which provides the only means of testing how well climate models are accurate for times of different planetary energy balances and different Earth-surface characteristics.
Reworded

Line 60 "the site was not covered by ice during MIS 11 and indicates the maximum limit for ice extent at that time" I think it is safer to say "the site was not covered by an ice sheet during a significant fraction of MIS 11" as there is no duration in the authors' datings from the core.
reworded "for some time during MIS 11"

Figure 6. This is a very important figure. From the lithostrat column it "looks like" three primary units. I take from this that Unit 1, the thickest unit, is pretty uniform in the characteristics you are using. But the other units are much more distributed into different clusters, which suggests that there may be different depositional and re-working processes at work in the younger units, which complicates genetic interpretations.
Yes, we agree that there are different depositional processes at work and there is reworking but we are not sure how this complicates genetic interpretations or how to integrate this comment. We have integrated this variability in clustering to our discussion.

1. Discussion

Line 519 "The rich record of ice sheet history" I suggest deleting "rich". I see this as an important depositional record that reflects a complex deglacial history during MIS 11 that is difficult to constrain in terms age beyond the MIS 11 window, and also of processes that produced the sediment in the core segments.
deleted

It would be very helpful to go through this section unit-by-unit starting with Unit1, with the authors briefly summarizing their interpretation of the depositional environment of their 5 "stratigraphic units" in the subglacial sediment and the reasons for that interpretation. Details for their interpretation can follow in the sections under this heading.
We tried to do this in the ms, admittedly not successfully as all three reviews have suggested that we provide more detailed reasoning for our interpretation. We have extensively revised the discussion and reorganized the interpretation of the units to be in line with the stratigraphy. In addition to the new figures we believe that it improves comprehension of our work.

**5.1 Sub-glacial Core Stratigraphy: Synthesis of Physical, Chemical, and Mineralogic Observations**
You should impose an order on this section, which summarizes your interpretations of the genesis of each unit based on all the data you generated starting with the basal Unit 1 and work your interpretation upward. As it is the text goes back and forth between the units, and it is too much work for the casual reader to follow.
Thanks for this advice which is consistent with that of other reviewers. We will made these changes per response above.

Line 535: I was not convinced the unit 3 was a slump. Could it not have been a "basal dirty ice unit from the MIS 12 glacial cycle? The underlying Unit 2 is dirty basal ice. Why couldn't Unit 3 be the upper part of Unit 2 that liquified when Unit 2 was exposed at the surface during an interglacial, with the fluvial processes imposing a different order on your primary measured variables?
There are geochemical data in other "in preparation papers" (major element and cosmogenic) that, in addition to the physical stratigraphy, support unit 3 being a slump. With the geochemistry of the materials in unit 2 and 3 differing and sedimentological evidence that neither unit 2 or 3 is a fluvial deposit, the evidence we have does not agree with the reviewer's interpretation. In answer to this comment (shared between reviewers) we have now emphasized the presence of a clear discontinuity between units 2 and 3 and provided evidence of slump-like features in the new figure.

Figure 8 This is a very helpful figure. I suggest adding to the right hand box the overall environmental condition as is done in Unit 3. Unit 1: Glaciation, Unit 2 Deglaciation, Unit 3 early Interglaciation, Unit 4 Interglaciation, Unit 5 transition into the next glaciation?
This is an interesting idea but we have no evidence indicating Unit 5 is the transition to the next glaciation and other reviewers have suggested that we do less speculative interpretation.

**Key points** as I saw them, which is very close to section 5.2. I think it makes more sense to list the items under 5.2 in terms of the 5 units identified
We attempted to do this but realized that presenting the discussion about the events doesn't allow

us to fully discuss the transition aspects which are fundamental. This approach gave too much focus on the units themselves which the rest of the paper does already.

The sub-ice sediment consists of sediment deposited both when ice was over the site and when it was ice free. Under the rule of interpretation following the "lines of least astonishment", it seemed to me the sequence below is the most parsimonious interpretation of depositional events. While we appreciate the suggestion, geochemical, isotopic, and geochronologic data do not support the reviewer's re-assessment of the history represented by the sediment recovered at Camp Century and so we will not modify the manuscript in the way suggested. We detail our reasoning below.

1. The 5 units start with a basal till (Unit 1), overlain by sediment entrained in the basal ice of a deglaciating MIS 12 glacial advance across the site (Unit 2).
   The highly weathered nature of unit 1 and the IRSL and cosmogenic data (burial between 0.7 to 3.2 Ma, Christ et al. 2021) provide no evidence it was deposited during MIS 12. Data collected so far (luminesce age > 1.7 My) do not support reviewer's suggestion of a MIS 12 age for unit 1. Data we have so far (stable isotopic, pore water and sediment geochemistry) are not diagnostic of a basal ice source for unit 2.

2. Unit 3. Represents MIS 12 deglacial sediment that is likely derived from sediment carried with in the deglaciating MIS 12 ice, as well as reworking of dirty basal ice as the ice margin retreats inland, resulting in some sediment characteristic similar or very similar to the basal till of unit 1.
   See above. We have no evidence that sediment in unit 3 or the ice in unit 2 is from MIS 12. We agree that unit 3 and unit 1 are geochemically similar and stress that as we revise the ms.

3. Once the landscape has stabilized in MIS 11, fluvial sediment dominates the environment, and this process evolves with increasingly aggressive flow in your upper units that may suggest the onset of an advancing MIS 10 Greenland Ice Sheet?
   We agree that fluvial sediment dominates the upper part of the core with increasing energy. We are not comfortable making any interpretation of an MIS 10 advance based on a single core at a single place given the spatial and temporal variability of glacial sediment systems over short time and length scales.

---

## Referee Report (RR1)

**Response egusphere-2024-2194**

The authors have done an excellent job with the revisions, and I no longer have any outstanding concerns. I recognize and appreciate the thoughtful attention they gave to the reviewer feedback. The updated Figures 2 and 3 stand out as particularly well-crafted.

*Below are a few very minor grammatical/punctuation issues:*

**Line 304:** 4*Area_/ (π*major) → Replace asterisk with multiplication symbol

Replace hyphens with en-dash for ranges throughout manuscript (eg **Line 305-306:** 0-1 should be 0–1 )

**Line 309:** 1000 um2 → µ instead of u

**Line 312:**  ImageJ, and a Tukey-Kramer honestly significant difference

**Line 320:** "performed a PCA analysis" is redundant (i.e. Principial Component Analysis analysis")

**Line 326**  a k-means clustering → the article "a" is unnecessary

**Line 327:** capitalize "z score" since starting sentence

Describe scans as either Micro-Ct or µCT rather than switching between abbreviations. Is the distinction between CT and µCT (lines 360 to 370 for example) denoting an actual difference in resolution or is it referring to the same µCT scan?

**Figure 8 caption line 1:** K-means clustering → should be "clustering"

circularity z score=0.9", "depth z score =1.2 → ensure correct spacing around equal signs eg **Lines 546, 549, 551** etc

"subglacial" is standard phrasing in the literature rather than "sub-glacial" (note that both are used **in line 646**)

---

## Author Response (AR2)

RESPONSE TO REVIEWS (minor revision)

Responses are in blue

**Editor**
The three reviewers are very satisfied with the revision of your initial manuscript and thank you for this considerable effort, with a special mention for the improvement of figures 2 and 3.

Thank you

However, they do highlight certain points that I would ask you to correct or improve. I reproduce here the various points I ask you to address.

We have addressed each point in our response.

In particular, I share reviewer 3's concerns about the use of luminescence dates in the authors' reasoning, but these dates do not appear to be available in the publication, which is a serious problem that I sincerely ask you to tackle.

Those dates are now available in a completed ms thesis (https://doi.org/10.26076/fa75-82e1) that we cite in the revised ms. We did not see the reviewer's concern in the review available to us online.

**Reviewer #1:**
In their rebuttal, the authors write that they are now citing Larson et al. (2016) and Menzies and Reitner (2016), but it does not appear in the revised manuscript.

These are now cited. Our omission.

The authors were also keen to keep their interpretation of "slump" for Unit 3, although the three reviewers were questioning this interpretation. If one looks at the Glossary of Geology definition of "slump", one can read :"Slump: a landslide characterized by a shearing and rotary movement of a generally independent mass of rock or earth along a curved slip surface and about an axis parallel to the slope from which it descends, and by backward tilting of the mass with respect to that slope so that the slump surface often exhibits a reversed slope facing uphill." OR "The sliding-down of a mass of sediment shortly after its deposition on an underwater slope." The first definition

implies that you need to know the geometry of the movement, which is not the case here. The second definition implies the presence of a water body, such as a lake, but this seems very unlikely in this context, as the authors are describing subaerial processes (line 595). I guess the misunderstanding arises from the confusion between the process that is taking place and the sedimentary facies that is the result of that process. I suggest the authors should use a more generic term such as "mass wasting" that can form the sediment structure that is described in the paper.

We agree with the term mass movement and have made the change in the ms. Mass movement is more clear and less specific than the word "slump"

Also, the authors use "flowtill", but its definition is "A superglacial till that is modified and transported by plastic mass flow." I'm not sure this is the correct use of that term in this paper. Maybe the authors mean "a till affected by solifluction"?

We agree with the reviewer. Other datasets (some yet to be published) suggest strongly that the material in unit 3 is derived from unit 1. Because we cannot be sure that solifluction is the processed we have reworded from *flowtill* to *diamicton (likely till) transported by a mass movement.*

Minor comments:
In the very nice new figure 2, sample 1061-D5 seems to belong to both unit 1 and unit 2 in the a. part of the figure, but it looks like it only belongs to unit 2 in part b. of the figure. The text seems to infer that sample 1061-D5 belongs to unit 1 and unit 2. Maybe should you change the lower limit unit 2 upward in panel b. of figure 2?

This is correct. Thank you for catching. We revised the figure.

L363: invert the depths here and below, the lowest first and then the highest, i.e., 327-223 cm

Change made

L371: there are only 6 samples in unit 2.

We make the change to reflect the dual nature of the top sample so that it now reads: "The next *six and a half* samples (1061-D2 to *the lowest portion of* 1060-C4) comprise Unit 2"...

L530-532: "As depth is relevant in stratigraphy and interpretations of stratigraphic deposits, we elected to retain depth as a variable to include as much information as possible in the algorithm." I'm not sure about this. If authors want to determine stratigraphic units, depth is an important variable, but if the authors want to identify different groups of sediments based on their properties, then including the depth indeed forces samples to be grouped in the same group if they are close to each other in the stratigraphy (spatial autocorrelation, as mentioned by a reviewer).

Thanks for this comment which indicates that we need to better explain the integration of depth in our clustering approach.

It is true that the integration of depth in our approach introduces a bias toward sample positioning in relation to each other. However, in stratigraphy, depth information is an explicit observation and typically integrated into clustering. This approach, sometimes called "depth-constrained cluster analysis" is used routinely in the context of well log zonation to improve grouping with an adjacency constraint (Gill et al., 1993; Yabe et al., 2022)

Within the context of parametric independence, we therefore choose to integrate stratigraphic interdependency as a deterministic factor. The central tenet of this approach is that relative depth is an inherent control on stratigraphic properties. In other words, it embraces the reality that, at the scale of the core, a given (artificially defined) sample is part of a stratigraphic continuum. Removing this parameter or rendering it independent would effectively negate the underlaying principles of stratigraphic continuity and superposition. By integrating depth as a parameter in the clustering we integrate the fact that samples are stratigraphically interconnected.

To that effect, we have now added a better justification to the use of depth dependency in the clustering method description so that the text reads (L324):

"By integrating depth as a factor in the clustering, we follow traditional clustering stratigraphic models acknowledging depth-dependency as a deterministic factor (Gill et al., 1993;  Yabe et al., 2022)" The corresponding references are also added to the ms.

**Reviewer 2:**
Line 304: 4*Area_/ (π*major) -> Replace asterisk with multiplication symbol

Replace hyphens with en-dash for ranges throughout manuscript (eg Line 305-306: 0-1should be 0–1 )

We are hesitant to make these changes prior to copy-edit but have no objection for these to be changed to journal style at copy edit.

Line 309: 1000 um2 à µ instead of u

Change made

Line 312: ImageJ, and a Tukey-Kramer honestly significant difference

Change made

Line 320: "performed a PCA analysis" is redundant (i.e. Principial Component Analysis analysis")

Change made

Line 326 a k-means clustering à the article "a" is unnecessary

Change made

Line 327: capitalize "z score" since starting sentence

Change made

Describe scans as either Micro-Ct or µCT rather than switching between abbreviations. Is the distinction between CT and µCT (lines 360 to 370 for example) denoting an actual difference in resolution or is it referring to the same µCT scan?

CT scan in our usage is different than µCT. We have adopted µCT throughout.

Figure 8 caption line 1: K-means clustering à should be "clustering"
circularity z score=0.9", "depth z score =1.2 à ensure correct spacing around equal signs eg Lines 546, 549, 551 etc

Change made

"subglacial" is standard phrasing in the literature rather than "sub-glacial" (note that both are used in line 646)

We have standardized to subglacial.

**Review 3**

Reviewer 3 has some concern about dates used but not suitable for the publication.

We as authors do not see that concern in material made available to us but those dates are now available in a completed ms thesis that we cite in the revised ms.